# LEARNING TO (LEARN AT TEST TIME):
# RNNS WITH EXPRESSIVE HIDDEN STATES

## ABSTRACT

Self-attention performs well in long context but has quadratic complexity. Existing RNN layers have linear complexity, but their performance in long context is limited by the expressive power of their hidden state. Inspired by prior work (Schlag et al., 2021), we present a practical framework to instantiate sequence modeling layers with linear complexity and expressive hidden states. The key idea is to make the hidden state a machine learning model itself, and the update rule a step of self-supervised learning. Since the hidden state is updated by training even on test sequences, our layers are called *Test-Time Training (TTT) layers*. We consider two instantiations: TTT-Linear and TTT-MLP, whose hidden state is a linear model and a two-layer MLP respectively. We evaluate our instantiations at the scale of 125M to 1.3B parameters, comparing with a strong Transformer and Mamba, a modern RNN. Both TTT-Linear and TTT-MLP match or exceed the baselines. Similar to Transformer, they can keep reducing perplexity by conditioning on more tokens, while Mamba cannot after 16k context. With preliminary systems optimization, TTT-Linear is already faster than Transformer at 8k context and matches Mamba in wall-clock time. TTT-MLP still faces challenges in memory I/O, but shows larger potential in long context, pointing to a promising direction for future research.

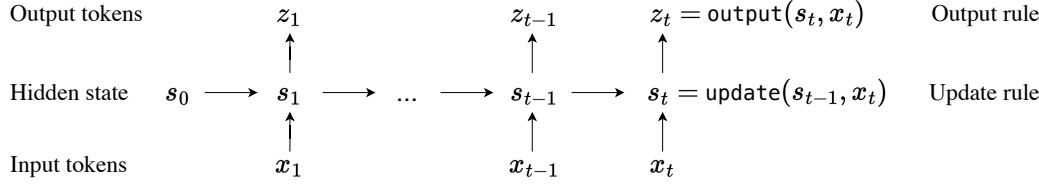

| | **Initial state** | **Update rule** | **Output rule** | **Cost** |
|---|---|---|---|---|
| **Naive RNN** | $s_0 = \texttt{vector()}$ | $s_t = \sigma\left(\theta_{ss}s_{t-1} + \theta_{sx}x_t\right)$ | $z_t = \theta_{zs}s_t + \theta_{zx}x_t$ | $O(1)$ |
| **Self-attention** | $s_0 = \texttt{list()}$ | $s_t = s_{t-1}.\texttt{append}(k_t, v_t)$ | $z_t = V_t\,\texttt{softmax}\left(K_t^T q_t\right)$ | $O(t)$ |
| **Naive TTT** | $W_0 = f.\texttt{params()}$ | $W_t = W_{t-1} - \eta\nabla\ell(W_{t-1}; x_t)$ | $z_t = f(x_t; W_t)$ | $O(1)$ |

Figure 1. **Top**: A generic sequence modeling layer expressed as a hidden state that transitions according to an update rule. All sequence modeling layers can be viewed as different instantiations of three components in this figure: the initial state, update rule and output rule. **Bottom**: Examples of sequence modeling layers and their instantiations of the three components. Self-attention has a hidden state growing with context, therefore growing cost per token. Both the naive RNN and TTT layer compress the growing context into a hidden state of fixed size, therefore their cost per token stays constant.

## 1 INTRODUCTION

Transformers have become the most popular architecture for large language models, but their cost per token grows linearly in context length. Recurrent Neural Networks (RNNs) have constant cost per token, but often struggle to actually express relationships in long context (Kaplan et al., 2020). This difficulty with long context is inherent to the very nature of RNN layers: Unlike self-attention, RNN layers have to compress context into a hidden state of fixed size. The update rule needs to discover the underlying structures and relationships among thousands or potentially millions of tokens. This is a very hard compression problem.

**TTT layers.** We first observe that self-supervised learning can compress a massive training set into the weights of a model. Motivated by this observation, we believe that sequence modeling layers can be designed with the hidden state as a machine learning model, and the update rule as a step of self-supervised learning. Because the process of updating the hidden state on a test sequence is equivalent to training a model at test time, we call these sequence modeling layers *Test-Time Training (TTT) layers*.

**Wall-clock time.** While the TTT layer is already efficient in FLOPs, we propose two practical innovations to make it efficient in wall-clock time. First, similar to the standard practice of taking gradient steps on mini-batches of sequences during regular training for better parallelism, we use mini-batches of tokens during TTT. Second, we develop a dual form for operations inside each TTT mini-batch, to better take advantage of modern GPUs and TPUs. The dual form is equivalent in output to the naive implementation, but trains more than $5\times$ faster.

**Contributions.** For self-containment, we discuss three levels of ideas in a single narrative: 1) Learning at test time and meta-learning at training time. 2) A practical framework to do the above with any neural network as inner model. 3) TTT-Linear and TTT-MLP as instantiations. Our contribution is only at level 2. Specifically, when the hidden state is a linear model, our design coincides with prior work (Schlag et al., 2021), which is further improved by concurrent work (Yang et al., 2024), as detailed in Subsection 2.6 and Subsection 4.2.

## 2 METHOD

All sequence modeling layers can be viewed from the perspective of storing historic context into a hidden state, as shown in Figure 1. For example, RNN layers – such as LSTM (Hochreiter & Schmidhuber, 1997), RWKV (Peng et al., 2024) and Mamba (Gu & Dao, 2023) layers – compress context into a state of fixed size across time. This compression has two consequences. On one hand, mapping an input token $x_t$ to output token $z_t$ is efficient, because both the update rule and output rule take constant time per token. On the other hand, the performance of RNN layers in long context is limited by the expressive power of its hidden state $s_t$.

Self-attention can also be viewed from the perspective above, except that its hidden state, commonly known as the Key-Value (KV) cache, is a list that grows linearly with $t$. Its update rule simply appends the current KV tuple to this list, and the output rule scans over all tuples up to $t$ to form the attention matrix. The hidden state explicitly stores all historic context, making self-attention more expressive than RNN layers for long context. However, scanning this linearly growing hidden state also takes linearly growing time per token.

To remain both efficient and expressive in long context, we need a better compression heuristic. Specifically, we need to compress thousands or potentially millions of tokens into a hidden state that can effectively capture their underlying structures and relationships.

### 2.1 TTT AS UPDATING A HIDDEN STATE

The process of parametric learning can be viewed as compressing a massive training set into the weights of a model. Specifically, we know that models trained with self-supervision can capture the underlying structures and relationships behind their training data (Le, 2013) – exactly what we need from a compression heuristic.

LLMs themselves are great examples. Trained with the self-supervised task of next-token prediction, their weights can be viewed as a compressed form of storage for existing knowledge on the internet. By querying LLMs, we can extract knowledge from their weights. More importantly, LLMs often exhibit a deep understanding of the semantic connections among existing knowledge to express new pieces of reasoning.

Our key idea is to use self-supervised learning to compress the historic context $x_1, \ldots, x_t$ into a hidden state $s_t$, by making the context an unlabeled dataset and the state a model. Concretely, the hidden state $s_t$ is now equivalent to $W_t$, the weights of a model $f$, which can be a linear model, a small neural network, or anything else. The output rule is simply: $z_t = f(x_t; W_t)$. Intuitively, the output token is just the prediction on $x_t$, made by $f$ with the updated weights $W_t$. The update rule is a step of gradient descent on some self-supervised loss $\ell$: $W_t = W_{t-1} - \eta \nabla \ell(W_{t-1}; x_t)$, with learning rate $\eta$. For now, consider $W_0 = 0$.

One choice of $\ell$ is reconstructing $x_t$ itself. To make the learning problem nontrivial, we first process $x_t$ into a corrupted input $\tilde{x}_t$ (details in Subsection 2.3), then optimize: $\ell(W; x_t) = \| f(\tilde{x}_t; W) - x_t \|^2$. Similar to denoising autoencoders (Vincent et al., 2008), $f$ needs to discover the correlations between dimensions of $x_t$ in order to reconstruct it from partial information $\tilde{x}_t$. As shown in Figure 7 in Appendix, gradient descent is able

to reduce $\ell$, but cannot reduce it to zero. We discuss more sophisticated formulations of the self-supervised task in Subsection 2.3.

As with other RNN layers and self-attention, our algorithm that maps an input sequence $x_1, \ldots, x_T$ to output sequence $z_1, \ldots, z_T$ can be programmed into the forward pass of a sequence modeling layer, using the hidden state, update rule, and output rule above. Even at test time, our layer still trains a different sequence of weights $W_1, \ldots, W_T$ for every input sequence. Therefore, we call it the *Test-Time Training (TTT) layer*.

## 2.2 TRAINING A NETWORK WITH TTT LAYERS

The forward pass of a TTT layer also has a corresponding backward pass. Our forward pass only consists of standard differentiable operators except the gradient operator $\nabla$. However, $\nabla$ just maps one function to another, in this case $\ell$ to $\nabla\ell$, and $\nabla\ell$ is also composed of differentiable operators. Conceptually, calling backward on $\nabla\ell$ means taking gradients of gradients – a well explored technique in meta-learning.

TTT layers have the same interface as RNN layers and self-attention, therefore can be replaced in any larger network architecture, which usually contains many of these sequence modeling layers. Training a network with TTT layers also works the same way as training any other language model, such as a Transformer. The same data, recipe, and objective such as next-token prediction can be used to optimize parameters of the rest of the network.

We refer to training the larger network as the *outer loop*, and training $W$ within each TTT layer as the *inner loop*. An important difference between the two nested learning problems is that the inner-loop gradient $\nabla\ell$ is taken w.r.t. $W$, the parameters of $f$, while the outer-loop gradient is taken w.r.t the parameters of the rest of the network, which we will denote by $\theta_{\text{rest}}$. So far, the TTT layer has no outer-loop parameters, in contrast to other RNN layers and self-attention.

## 2.3 LEARNING A SELF-SUPERVISED TASK FOR TTT

Arguably the most important part of TTT is the self-supervised task, because it determines the kind of features that $W$ will learn from the test sequence. So how should we design this task? The final goal of TTT is for $z_t = f(x_t; W_t)$ to perform well on language modeling. Instead of handcrafting a self-supervised task from human priors, we take a more end-to-end approach – directly optimizing the self-supervised task for the final goal of next-token prediction.

Concretely, we learn the self-supervised task as part of the outer loop. Starting from the naive reconstruction task in Equation 2.1, we add some outer-loop parameters to make this task learnable. In Subsection 2.1, we did not specify the corruption that produces $\tilde{x}_t$ from $x_t$. One design is to make it a low-rank projection $\tilde{x}_t = \theta_K x_t$, where $\theta_K$ is a learnable matrix. Following the terminology of multi-view reconstruction, $\theta_K x_t$ is called a *training view* (Chen et al., 2020).

Moreover, perhaps not all the information in $x_t$ is worth remembering, so the reconstruction label can be another low-rank projection $\theta_V x_t$ instead of $x_t$. Here $\theta_V x_t$ is called the *label view*, where $\theta_V$ is also learnable. In summary, our new self-supervised loss is: $\ell(W; x_t) = \|f(\theta_K x_t; W) - \theta_V x_t\|^2$. Lastly, the training view $\theta_K x_t$ has fewer dimensions than $x_t$, so we can no longer use the output rule in Equation 2.1. The simplest solution is to create a *test view* $\theta_Q x_t$, and change our output rule to: $z_t = f(\theta_Q x_t; W_t)$.

## 2.4 PARALLELIZATION WITH MINI-BATCH TTT

The naive TTT layer developed so far is already efficient in the number of floating point operations (FLOPs). However, its update rule $W_t = W_{t-1} - \eta\nabla l(W_{t-1}; x_t)$ cannot be parallelized, because $W_t$ depends on $W_{t-1}$ in two places: before the minus sign and inside $\nabla l$. Since $\nabla l$ contains the bulk of the computation, we focus on making this second part parallel.

We approach this systems challenge through concepts in the TTT framework. There are many variants of gradient descent (GD). The general update rule of GD can be expressed as:

$$W_t = W_{t-1} - \eta\, G_t = W_0 - \eta \sum_{s=1}^{t} G_s, \tag{1}$$

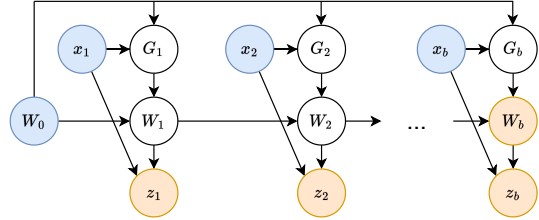

Figure 2. High-level computation graph of the first TTT mini-batch, where nodes are variables and edges are computations. The blue nodes are input variables, and yellow are output. **Subsection 2.4**: Since $G_1, \ldots, G_b$ are not connected, they have no sequential dependency on each other, therefore can be computed in parallel. **Subsection 2.5**: We do not actually materialize the white nodes – the intermediate $G$s and $W$s – to compute the output variables in the dual form.

where $G_t$ is the descent direction. Note that once we have calculated $G_t$ for $t = 1, \ldots, T$, we can then obtain all the $W_t$s through a cumsum by the second half of Equation 1. Our naive update rule, known as *online gradient descent*, uses $G_t = \nabla l(W_{t-1}; x_t)$.

To parallelize $G_t$ for $t = 1, \ldots, T$, we can take all of them w.r.t. $W_0$. This variant with $G_t = \nabla \ell (W_0; x_t)$ is known as *batch gradient descent*, since $\sum_{s=1}^{t} \nabla \ell (W_0; x_s)$ is the same as the gradient w.r.t. $W_0$ over $x_1, \ldots, x_t$ as a batch. However, in batch GD, $W_t$ is effectively only one gradient step away from $W_0$, in contrast to online GD, where $W_t$ is $t$ steps away from $W_0$. Therefore, batch GD has a smaller effective search space, which ends up hurting performance for language modeling.

Our proposed solution – *mini-batch gradient descent* – is shown in Figure 2. Denote the TTT batch size by $b$. We use $G_t = \nabla \ell (W_{t'}; x_t)$, where $t' = t - \text{mod}(t, b)$ is the last timestep of the previous mini-batch (or 0 for the first mini-batch), so we can parallelize $b$ gradient computations at a time. Empirically, $b$ controls a trade-off between speed and quality, as shown in Figure 9. We chose $b = 16$ for all experiments in this paper.

## 2.5 DUAL FORM

The parallelization introduced above is necessary but not sufficient for efficiency in wall-clock time. Modern accelerators specialize in matrix-matrix multiplications, known as matmuls. Unfortunately, the TTT layer developed so far even with mini-batch still has very few matmuls.

Consider the simplest case of $\ell$, where $\theta_K = \theta_V = \theta_Q = I$, for only the first TTT mini-batch of size $b$. In addition, consider $f$ as a linear model. Copying Equation 2.1, our loss at time $t$ is: $\ell (W_0; x_t) = \|W_0 x_t - x_t\|^2$. As discussed in Subsection 2.4, we can parallelize the computation of: $G_t = \nabla \ell (W_0; x_t) = 2(W_0 x_t - x_t)x_t^T$, for $t = 1, \ldots, b$. However, we cannot compute all $b$ of the $G_t$s through a single matmul. Instead, we need $b$ outer products to compute them one by one. To make matters worse, for each $x_t \in \mathbb{R}^d$, $G_t$ is $d \times d$, which incurs much heavier memory footprint and I/O cost than $x_t$ for large $d$.

To solve these two problems, we make a simple observation: We do not actually need to materialize $G_1, \ldots, G_b$ as long as we can compute $W_b$ at the end of the mini-batch, and the output tokens $z_1, \ldots, z_b$ (see Figure 2). Now we demonstrate these computations with the simplified TTT-Linear case above. Denote $X = [x_1, \ldots, x_b]$,

$$W_b = W_0 - \eta \sum_{t=1}^{b} G_t = W_0 - 2\eta \sum_{t=1}^{b} (W_0 x_t - x_t)x_t^T = W_0 - 2\eta(W_0 X - X)X^T.$$

So $W_b$ can be conveniently computed with a matmul. To compute $Z = [z_1, \ldots, z_b]$, we know that:

$$z_t = f(x_t; W_t) = W_t x_t = \left( W_0 - \eta \sum_{s=1}^{t} G_t \right) x_t = W_0 x_t - 2\eta \sum_{s=1}^{t} (W_0 x_s - x_s)x_s^T x_t. \quad (2)$$

Denote $\delta_t = \sum_{s=1}^{t} (W_0 x_s - x_s)x_s^T x_t$ and the matrix $\Delta = [\delta_1, \ldots, \delta_b]$, then $\Delta = (W_0 X - X) \, \text{mask} \left( X^T X \right)$, where mask is the upper triangular mask with zeros, and the term $W_0 X - X$ can be reused from the computation of $W_b$. Now $\Delta$ is also computed with matmuls. Plugging $\Delta$ back into Equation 2, we obtain $Z = W_0 X - 2\eta\Delta$. We call this procedure the *dual form*, in contrast to the *primal form* before this subsection, where the $G$s and $W$s are explicitly materialized. As discussed, the two forms are equivalent in output. In Appendix A, we show that the dual form still works when $f$ is a neural network with nonlinear layers.

Time complexity of the primal form within a TTT mini-batch is $O(b \times d^2)$. Time complexity of the dual form is $O(b \times d^2)$ for computing $W_b$ alone, then an additional $O(b^2 \times d)$ for computing $z_1, \ldots, z_b$. Compared to the primal, the dual form sacrifices theoretical complexity for hardware utilization. In practice, $d$ is typically a few hundred and $b$ is chosen to be only 16. As a consequence, wall-clock time for computing $z_1, \ldots, z_b$ is relatively small, as observed in the right panel of Figure 9 in Appendix.

| Configuration | Ppl. | Diff. |
|---|---|---|
| Linear attention | 15.91 | - |
| Linear attn. improved | 15.23 | $-0.68$ |
| TTT equivalence | 15.23 | 0 |
| + learnable $W_0$ | 15.27 | $+0.04$ |
| + LN and residual in $f$ | 14.05 | $-1.22$ |
| + mini-batch TTT | 12.35 | $-1.70$ |
| + learnable $\eta$ | 11.99 | $-0.36$ |
| + Mamba backbone | 11.09 | $-0.90$ |

Table 1. Ablations on improving from linear attention. All models here have 125M parameters, and are trained according to the recipe in Subsection 3.1. The last row, with perplexity 11.09, is the final result of TTT-Linear in Figure 3. Starting from the equivalence discussed in Subsection 2.6, learnable $W_0$ hurts slightly, but the rows below cannot train stably without it. The biggest improvement comes from mini-batch TTT (changing from $b = T = 2048$ to $b = 16$). The second comes from instantiating the inner model $f$ with LN and residual connection. Both of these designs would be difficult to come across without the conceptual framework of TTT.

### 2.6 THEORETICAL EQUIVALENCES

In Subsection 2.1, we mentioned that $f$ can be a linear model or a neural network. In Subsection 2.4, we also discussed three variants of the update rule: online GD, batch GD, and mini-batch GD. Each of these $2 \times 3$ combinations induces a different instantiation of the TTT layer. We now show that among these induced instantiations, the TTT layer with a linear model and batch GD is equivalent to linear attention (Katharopoulos et al., 2020), a widely known RNN layer.

**Theorem 1.** *Consider the TTT layer with $f(x) = Wx$ as the inner-loop model, batch gradient descent with $\eta = 1/2$ as the update rule, and $W_0 = 0$. Then, given the same input sequence $x_1, \ldots, x_T$, the output rule defined in Equation 2.3 produces the same output sequence $z_1, \ldots, z_T$ as linear attention.*

*Proof.* $\nabla \ell (W_0; x_t) = -2(\theta_V x_t)(\theta_K x_t)^T$, so $W_t = W_{t-1} - \eta \nabla \ell (W_0; x_t) = W_0 - \eta \sum_{s=1}^{t} \nabla \ell (W_0; x_s) = \sum_{s=1}^{t} (\theta_V x_s)(\theta_K x_s)^T$. Plugging $W_t$ into the output rule in Equation 2.3, we obtain the output token:

$$z_t = f(\theta_Q x_t; W_t) = \sum_{s=1}^{t} (\theta_V x_s)(\theta_K x_s)^T (\theta_Q x_t),$$

which is the definition of linear attention. $\square$

In Table 1, we first empirically verify the equivalence above with an improved implementation of linear attention. Then, to illustrate the contribution of each of our components (including some that will be introduced in the next subsection), we add them row by row to the TTT layer that is equivalent to linear attention, and ultimately obtain our proposed instantiation called *TTT-Linear*. The change from batch GD to mini-batch GD contributes the most improvement by a large margin.

While the space of models $\times$ optimizers is already large, machine learning is much richer than optimizing the parameters $W_t$ of a model $f$. There are also nonparametric learners, such as nearest neighbors, support vector machines (SVMs), and kernel ridge regression. By definition, nonparametric learners do not have parameters $W_t$, and instead directly uses training data $x_1, \ldots, x_t$. Hence we use the notation $f(x; x_1, \ldots, x_t)$. We now show that for a particular nonparametric learner, the induced TTT layer is equivalent to self-attention.

**Theorem 2.** *Consider the TTT layer with the Nadaraya-Watson estimator (Bierens, 1988), defined as:*

$$f(x; x_1, \ldots, x_t) = \frac{1}{\sum_{s=1}^{t} \kappa(x, x_s)} \sum_{s=1}^{t} \kappa(x, x_s) \, y_s, \tag{3}$$

*where $y_s = \theta_V x_s$ is the label view discussed in Subsection 2.3, and $\kappa (x, x'; \theta_K, \theta_Q) \propto e^{(\theta_K x)^T \theta_Q x'}$ is a kernel with bandwidth hyper-parameters $\theta_K$ and $\theta_Q$. Then given the same input sequence $x_1, \ldots, x_T$, the output rule defined in Equation 2.3 produces the same output sequence $z_1, \ldots, z_T$ as self-attention.*

*Proof.* Plugging $y_s$ and $\kappa$ above into Equation 3 gives us the definition of self-attention. Appendix B contains a detailed explanation of the Nadaraya-Watson estimator and kernel $\kappa$ above. $\square$

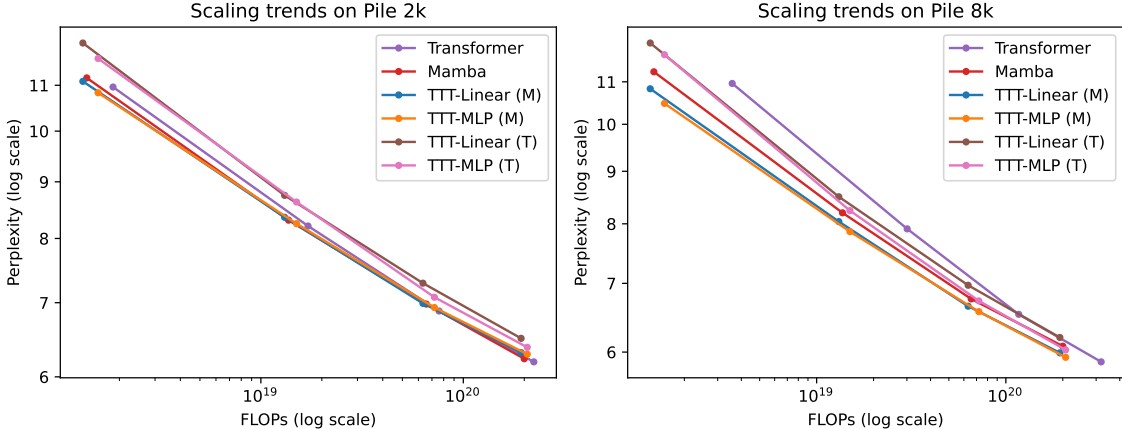

Figure 3. Evaluations for context lengths 2k and 8k on the Pile. Details in Subsection 3.1. TTT-Linear has comparable performance as Mamba at 2k context, and better performance at 8k.

## 2.7 IMPLEMENTATION DETAILS

**Instantiations of $f$.** We propose two variants of TTT layers – TTT-Linear and TTT-MLP, differing only in their instantiations of $f$. For TTT-Linear, $f_{\texttt{lin}}(x) = Wx$, where $W$ is square. For TTT-MLP, $f_{\texttt{MLP}}$ has two layers similar to the MLPs in Transformers. Specifically, the hidden dimension is $4\times$ the input dimension, followed by a GELU activation (Hendrycks & Gimpel, 2016). For better stability during TTT, $f$ always contains a Layer Normalization (LN) and residual connection. That is, $f(x) = x + \texttt{LN}(f_{\texttt{res}}(x))$, where $f_{\texttt{res}}$ can be $f_{\texttt{lin}}$ or $f_{\texttt{MLP}}$.

**Learnable $W_0$.** The TTT initialization $W_0$ is shared between all sequences, even though subsequent weights $W_1, \ldots, W_T$ are different for each input sequence. Instead of setting $W_0 = 0$, we can learn it as part of the outer loop. Since outer-loop parameters are always denoted by $\theta$s instead of $W$s, we assign an alias $\theta_{\text{init}} = W_0$. In practice, $\theta_{\text{init}}$ adds a negligible amount of parameters comparing to the reconstruction views $\theta_K, \theta_Q, \theta_V$, because both its input and output are low dimensional. Empirically, we observe that learning $W_0$ significantly improves training stability.

**Learnable $\eta$.** The learning rate is usually the most important hyper-parameter for gradient descent, so we experiment with learning the inner-loop learning rate $\eta$ in Equation 1 as part of the outer loop. We make $\eta$ a function of the input token (therefore different across time) for additional flexibility. Concretely, we design $\eta(x) = \eta_{\text{base}} \sigma(\theta_{\text{lr}} \cdot x)$, where the learnable vector $\theta_{\text{lr}}$ is an outer-loop parameter, $\sigma$ is the sigmoid function, and the scalar $\eta_{\text{base}}$ is the base learning rate, set to 1 for TTT-Linear and 0.1 for TTT-MLP. Alternatively, $\eta(x)$ can also be interpreted as a gate for $\nabla \ell$.

**Backbone architecture.** The cleanest way to integrate any RNN layer into a larger architecture would be to directly replace self-attention in a Transformer, known in this context as a backbone. However, existing RNNs such as Mamba and Griffin all use a different backbone from Transformers. Most notably, their backbone contains temporal convolutions before the RNN layers, which might help collect local information across time. After experimenting with the Mamba backbone, we find that it also improves perplexity for TTT layers, so we incorporate it into our proposed method. See Figure 10 (in Appendix) for details.

## 3 EXPERIMENTS

Our main codebase is based on EasyLM (Geng, 2023), an open-source project for training and serving LLMs in JAX. We evaluate TTT-Linear and TTT-MLP by comparing with Transformer and Mamba. As discussed in Subsection 2.7, Transformer and Mamba use different backbones, and TTT-Linear and TTT-MLP always use the Mamba backbone unless noted otherwise. As an ablation study, Figure 3 and Figure 4 contain TTT layers within the Transformer backbone. When a figure contains both the Transformer backbone and Mamba backbone, we denote them by *(T)* and *(M)*, respectively.

**Datasets.** Following the Mamba paper (Gu & Dao, 2023), we perform standard experiments with 2k and 8k context lengths on the Pile (Gao et al., 2020), a popular dataset of documents for training open-source

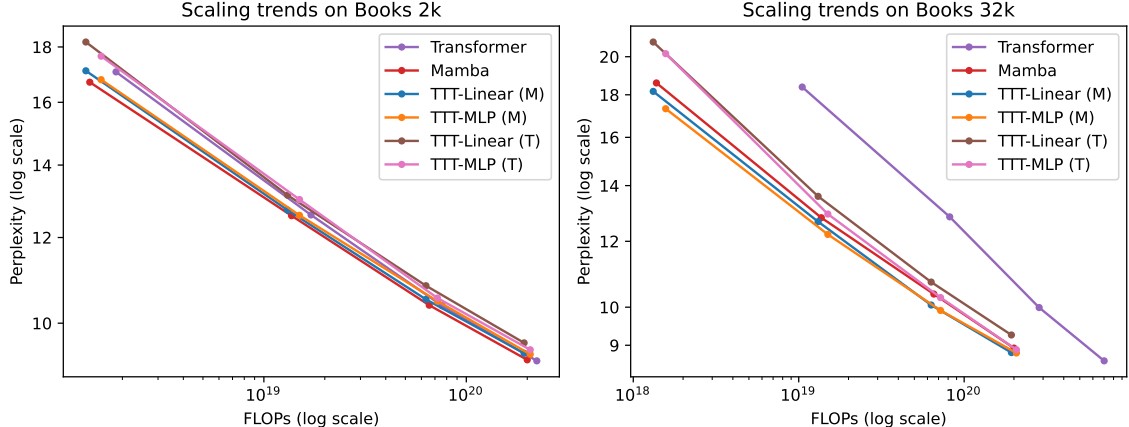

Figure 4. Evaluations for context lengths 2k and 32k on Books. Details in Subsection 3.2. Our complete results for context lengths 1k, 2k, 4k, 8k, 16k, 32k, including Transformer finetuning, are in Figure 11 (in Appendix). Most observations from the Pile still hold.

LLMs (Black et al., 2022). However, the Pile contains few sequences of length greater than 8k (de Vries, 2023). To evaluate capabilities in long context, we also experiment with context lengths ranging from 1k to 32k in $2\times$ increments, on a subset of the Pile called Books3, which has been widely used to train LLMs in long context (Liu et al., 2024; Authors Guild, 2023).

**Protocols.** To ensure fairness to our baselines, we strictly follow the evaluation protocols in the Mamba paper when possible. For each evaluation setting (e.g., dataset, context length, and method), we experiment with four model sizes: 125M, 350M, 760M, and 1.3B parameters. For Mamba, the corresponding sizes are 130M, 370M, 790M, and 1.4B, as Mamba does not follow the Transformer configurations. All models are trained with the Chinchilla recipe described in the Mamba paper. Our Transformer baseline, based on the Llama architecture (Touvron et al., 2023), also follows the baseline in the Mamba paper.

### 3.1 SHORT CONTEXT: THE PILE

From Figure 3, we make a few observations:

- At 2k context, TTT-Linear (M), Mamba, and Transformer have comparable performance, as the lines mostly overlap. TTT-MLP (M) performs slightly worse under large FLOP budgets. Even though TTT-MLP has better perplexity than TTT-Linear at every model size, the extra cost in FLOPs offsets the advantage.
- At 8k context, both TTT-Linear (M) and TTT-MLP (M) perform significantly better than Mamba, in contrast to the observation at 2k. Even TTT-MLP (T) with the Transformer backbone performs slightly better than Mamba around 1.3B. A robust phenomenon we observe throughout this paper is that as context length grows longer, the advantage of TTT layers over Mamba widens.
- At 8k context, Transformer still has good (if not the best) perplexity at every model size, but its line is not competitive because of the cost in FLOPs.

**Effect of backbone.** Switching the TTT layers from Mamba backbone into Transformer backbone has two effects. First, TTT layers with Mamba backbone perform better in our evaluations so far. Second, with Mamba backbone, TTT-MLP at best is only comparable to TTT-Linear; but with Transformer backbone, TTT-MLP is clearly better. We hypothesize that the temporal convolutions in the Mamba backbone help more when the sequence modeling layer has a less expressive hidden state. The linear model is less expressive than the MLP, therefore benefits more from the convolutions. We will revisit this hypothesis in the next subsection.

### 3.2 LONG CONTEXT: BOOKS

To evaluate capabilities in long context, we experiment with context lengths ranging from 1k to 32k in $2\times$ increments, using a popular subset of the Pile called Books3. The training recipe here is the same as for the Pile, and all experiments for the TTT layers are performed in one training run. From the subset of results in Figure 4, we make a few observations:

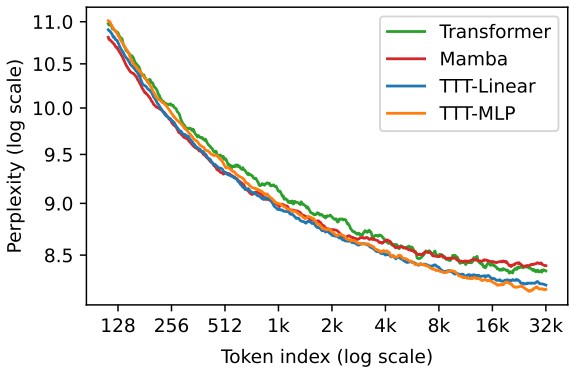

Figure 5. Evaluations following Kaplan et al. (2020). Tokens later in a sequence should be easier to predict on average, since they condition on longer context. Transformer, TTT-Linear and TTT-MLP can keep reducing perplexity as they condition on more tokens, while Mamba cannot after 16k context. Comparing to TTT-Linear, TTT-MLP performs slightly worse at short context but better at long context. This observation matches our expectation that the MLP as hidden state is more expressive than the linear model. All methods have matched training FLOPs as Mamba.

- At 2k context on Books, all the observations from Pile 2k still hold, except that Mamba now performs slightly better than TTT-Linear (whereas their lines roughly overlapped for Pile 2k).
- At 32k context, both TTT-Linear (M) and TTT-MLP (M) perform better than Mamba, similar to the observation from Pile 8k. Even TTT-MLP (T) with the Transformer backbone performs slightly better than Mamba at 32k context.
- TTT-MLP (T) is only slightly worse than TTT-MLP (M) at 1.3B scale. The strong trend for TTT-MLP (T) suggests that the Transformer backbone might be more suitable for larger models and longer context beyond our evaluations. We only ablate the backbones for 2k and 32k due to the cost of training LLMs. For future work, we believe that given TTT layers with even more expressive hidden states, the Mamba backbone with temporal convolutions will become unnecessary.

**Transformer finetuning.** While we have been training Transformers from scratch following the Mamba paper, in practice this approach is rarely used for long context. The standard practice is to train a Transformer in short context, then finetune in long context. To reflect this practice, we add another baseline, *TF finetune*, for context lengths 4k and above. This baseline starts from the model trained (according to the Chinchilla recipe) on Books 2k, then uses 20% more tokens to finetune at the designated context length, following the Llama Long paper (Xiong et al., 2023). See details of the TF finetune recipe in Appendix C.

Our complete results for context lengths 1k to 32k, including TF finetune, are in Figure 11 (in Appendix). TF finetune performs significantly better than TF pretrain, as it benefits from long context without incurring extremely large cost in training FLOPs. Note that the inference FLOPs of TF finetune and pretrain are equally poor, which is not reflected in this plot.

### 3.3 WALL-CLOCK TIME

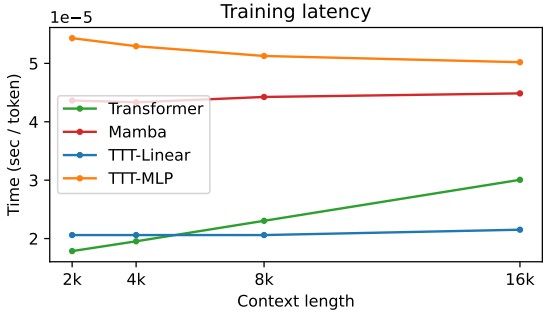

Figure 6. Benchmark on an NVIDIA H100 GPU. All models are 1.3B. Time per token grows linearly for Transformer as context length increases, but stays roughly constant for the other methods. TTT-Linear model is 2x faster than Mamba, and TTT-MLP is 20% slower than Mamba. Note that our Transformer baseline is significantly faster then that in the Mamba paper, because we use vLLM (Kwon et al., 2023), a state-of-the-art system, instead of the HuggingFace Transformer (Wolf et al., 2019).

LLM training and inference can be decomposed into forward, backward, and generate. Prompt processing during inference, also known as prefill, is the same operation as forward during training, except that the intermediate activations do not need to be stored for backward. Since both forward (during training and inference) and backward can be parallelized, we use the dual form. Generating new tokens, also known as decode, is inherently sequential, so we use the primal form.

We write a GPU kernel for forward and backward in ThunderKittens (Spector et al., 2023), and for generate in Triton (Tillet et al., 2019). Figure 6 shows the latency for training. Figure 13 in Appendix shows the latency for prefill and decode.

## 4 RELATED WORK

### 4.1 MODERN RNNs

Mamba is one of the many Structured State-Space Models (Gu et al., 2021; Fu et al., 2022; Poli et al., 2023; De et al., 2024). The hidden state in these models is a vector, similar to in LSTMs. For TTT-Linear or TTT-MLP, the hidden state is a matrix or two matrices, therefore larger. In Figure 5, we find that TTT layers can take advantage of their larger hidden states to compress more information in long context, where TTT-MLP outperforms TTT-Linear, which in turn outperforms Mamba.

Similar to TTT-Linear, RWKV (Peng et al., 2023; 2024), xLSTM (Beck et al., 2024), and Gated Linear Attention (GLA) (Yang et al., 2023) also have matrix hidden states, which are inherited from linear attention (Katharopoulos et al., 2020). Modern RNNs such as GLA use chunk-wise parallelism to improve hardware efficiency, so tokens inside a chunk can be processed with `matmuls` instead of a `cumsum`. However, chunk-wise parallelism does not change the expressiveness of the model, since all temporal dependencies are still equivalent to a `cumsum`.

In contrast, mini-batch TTT allows more complex temporal dependencies across mini-batches. Each hidden state $W_t$ depends on previous $W$s within its mini-batch still through a `cumsum`, but depends on $W$s in previous mini-batches also through the gradient operator. As illustrated Figure 9 in Appendix, mini-batch TTT enables a trade-off between expressiveness and hardware efficiency, since a smaller batch size $b$ leads to better perplexity at the cost of higher latency. This trade-off is a unique and important feature of TTT. As shown in Table 1, the intermediate batch size $b = 16$ significantly outperforms $b = T$ which is fully `cumsum`.

Concurrent work Mamba 2 (Dao & Gu, 2024) is similar to linear attention and TTT-Linear in that it also uses matrix hidden states. In fact, Mamba 2 is equivalent in output to linear attention with explicit forget gates (the scalars $a_t$ in their dynamical system) and a different backbone (which is also different from the original Mamba backbone). Conceptually, the improvements from Mamba on linear attention are orthogonal to those from TTT-Linear (e.g. mini-batch and LN, as shown in Table 1), and can potentially be combined to produce even stronger results.

TTT-Linear is also similar to DeltaNet (Yang et al., 2024), another piece of concurrent work. In fact, if we take away non-linearities such as LN and set inner-loop mini-batch size $b = 1$ instead of 16, a TTT-Linear layer is exactly equivalent to the sequence modeling layer in DeltaNet, and the update rule is known as the Delta rule (Schlag et al., 2020). Yang et al. (2024) takes advantage of this linearity and designs an alternative parallelization that is highly efficient. Comparing to DeltaNet, our method can instantiate any neural network as inner model and still maintain reasonable efficiency.

### 4.2 LEARNING AT TEST TIME

The idea of learning at test time has a long history in machine learning (Bottou & Vapnik, 1992; Gammerman et al., 1998; Sun et al., 2020). More recently, the same idea has also been applied to natural language processing, where it is called dynamic evaluation (Krause et al., 2018; 2019). The basic approach is to directly finetune a language model on the test sequence, which often comes in the form of a prompt. Following the same spirit as dynamic evaluation, Yoshida & Gimpel (2021) optimizes the next-token prediction loss of the test sequence with respect to the entire KV cache of a Transformer.

Next we discuss an especially relevant line of work in detail: fast weights. The general idea is to update the parameters of a "fast" model on only the most relevant data, as opposed to the conventional practice of updating a "slow" model on all data (Tieleman & Hinton, 2009). This idea has existed since the 1980s (Hinton & Plaut, 1987). TTT can be viewed as a special case of fast weights.

Prior work in fast weights usually avoids forming an explicit learning problem that optimizes some objective on data. For example, the update rule of Hebbian learning and Hopfield networks (Hopfield, 1982) simply adds $xx^T$ (or some variant thereof) (Ba et al., 2016) to the fast weights given each input $x$. In contrast,

TTT embraces the idea of formulating an explicit learning problem, where the test instance is the target of generalization. Our update rule is also an explicit step of optimization.

The idea of *fast weight programmers* (FWPs) is to update the fast weights with a "slow" model (Schmidhuber, 1992). Our inner-loop weights $W$ can be viewed as "fast" and outer-loop weights $\theta$ as "slow". Therefore, networks containing TTT layers can be viewed as a special case of FWPs (Kirsch & Schmidhuber, 2021), similar to how TTT can be viewed as a special case of fast weights. The FWP with the Hebbian update rule above is equivalent to linear attention (Schlag et al., 2021), therefore also to naive TTT-Linear with batch gradient descent.

The definition of FWPs is very broad. In fact, all networks with some gating mechanism, such as Transformers with SwiGLU blocks (Shazeer, 2020), can also be viewed as a special case of FWPs. Recent work has been experimenting with FWPs for language modeling: Irie et al. (Irie et al., 2021) design "fast" networks with weights produced as output of a "slow" networks. Clark et al. (Clark et al., 2022) give a Transformer a final layer of fast weights, whose initialization is trained as slow weights. Our contribution relative to existing work on FWPs, again, is formulating an explicit learning problem for the update, which enables us to borrow tools from learning such as mini-batch and LN.

### 4.3 Learning to Learn

For decades, researchers have been arguing that learning to learn, also known as meta-learning or bi-level optimization, should be a critical component of intelligence Schmidhuber (1987); Bengio et al. (1990); Thrun & Pratt (1998); Lake et al. (2017). In prior work such as Andrychowicz et al. (2016), Finn et al. (2017) and Metz et al. (2018), the inner loop learns from an entire dataset at a time instead of a sequence, so the outer loop needs a collection of datasets or tasks. In short, the outer loop is "one level above" regular training. Since it is hard to collect millions of datasets, this outer loop is hard to scale.

In contrast, for TTT, each sequence itself is a dataset and defines its own generalization problem. The inner loop is "one level below" regular training, so our outer loop is only another solution to the canonical problem of supervised learning, instead of a new problem setting like generalization across datasets. Our outer loop is "at the same level" as regular training. This makes our outer loop easier to scale.

## 5 Conclusion

We have reformulated the canonical problem of supervised learning as learning to (learn at test time). Our formulation produces an alternative conceptual framework for building what is traditionally known as network architectures. Our next goals are longer context, larger models, and more ambitious inner models. Next we outline some especially promising directions for future work.

- **Outer-loop parameterization.** There are many other ways to parameterize a family of multi-view reconstruction tasks, or perhaps a more general family of self-supervised tasks. It would be a big coincidence if the first one we have tried turns out to be the best.

- **Systems optimization.** Our systems optimization in Subsection 3.3 has been preliminary at best, and there are many ways to improve it. In addition, pipeline parallelism through time might allow us to process long sequences of millions of tokens on multiple devices together.

- **Longer context and larger models.** Constrained by our academic resources, we have not trained with millions or billions in context length, which would also require larger models according to Figure 12. The advantage of TTT layers should become more pronounced in longer context.

- **More ambitious instantiations of $f$.** When context length becomes longer, $f$ would also need to be larger. For video tasks and embodied agents, whose context length can easily scale up to millions or billions, $f$ could be a convolutional neural network.

- **Multi-level learning to learn.** If $f$ itself is a self-attention layer, then by Theorem 2 it can be interpreted as yet another inner loop nested inside the existing one. In this fashion, we can potentially build many levels of nested learning problems.

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

APPENDIX

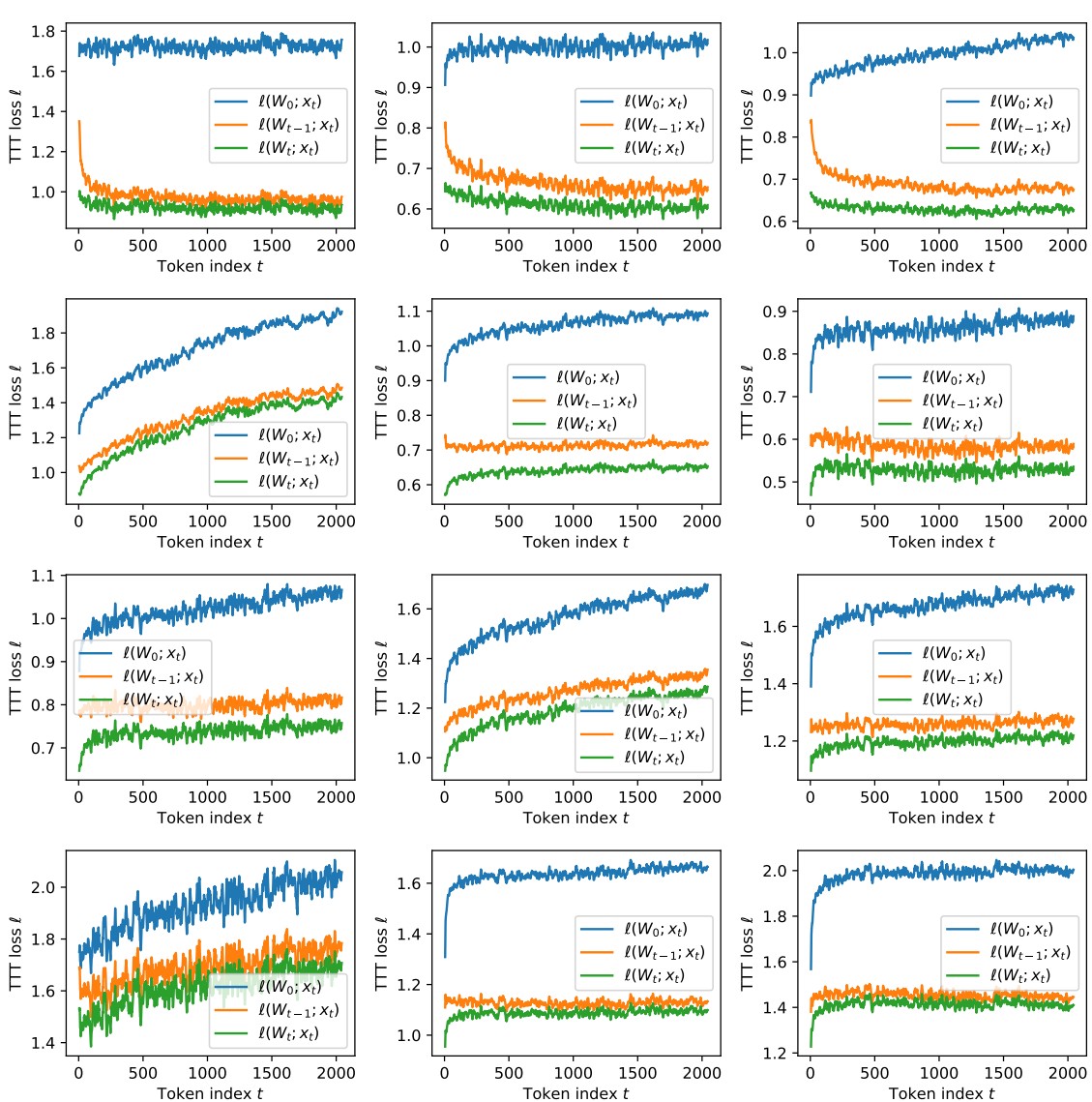

Figure 7. The self-supervised TTT loss $\ell$ averaged over all test sequences of the form $x_1, \ldots, x_T$ where $T = 2048$, for all 12 TTT layers in a network with 125M parameters train on the Pile. The same network is also used for $b = 1$ (online GD) in the left panel of Figure 9. For layers in the middle, we observe that $\|x_t\|$ rises steadily, causing all three losses to rise with it. Even for these layers, the gap between $\ell(W_0; x_t)$ and $\ell(W_t; x_t)$ still increases with $t$. For visual clarity, loss values have been averaged over a sliding window of 10 timesteps.

```python
class TTT_Layer(nn.Module):                    class Learner():
  def __init__(self):                            def __init__(self, task):
    self.task = Task()                             self.task = task
                                                   # Linear here, but can be any model
  def forward(self, in_seq):                       self.model = Linear()
    state = Learner(self.task)                     # online GD here for simplicity
    out_seq = []                                   self.optim = OGD()
    for tok in in_seq:
      state.train(tok)                           def train(self, x):
      out_seq.append(state.predict(tok))           # grad function wrt first arg
    return out_seq                                  # of loss, which is self.model
                                                    grad_fn = grad(self.task.loss)
class Task(nn.Module):                              # calculate inner-loop grad
  def __init__(self):                               grad_in = grad_fn(self.model, x)
    self.theta_K = nn.Param((d1, d2))
    self.theta_V = nn.Param((d1, d2))              # starting from current params,
    self.theta_Q = nn.Param((d1, d2))              # step in direction of grad_in,
                                                    self.optim.step(self.model, grad_in)
  def loss(self, f, x):
    train_view = self.theta_K @ x               def predict(self, x):
    label_view = self.theta_V @ x                 test_view = self.task.theta_Q @ x
    return MSE(f(train_view), label_view)         return self.model(test_view)
```

Figure 8. Naive implementation of a TTT layer with a linear model and online GD in the style of PyTorch. TTT_Layer can be dropped into a larger network like other sequence modeling layers. Training the network will optimize the parameters of Task in TTT_Layer, because both are subclasses of nn.Module. Since Learner is not a subclass of nn.Module, state.model is updated manually in the inner loop for each call of state.train. For simplicity, we sometimes overload model as model.parameters.

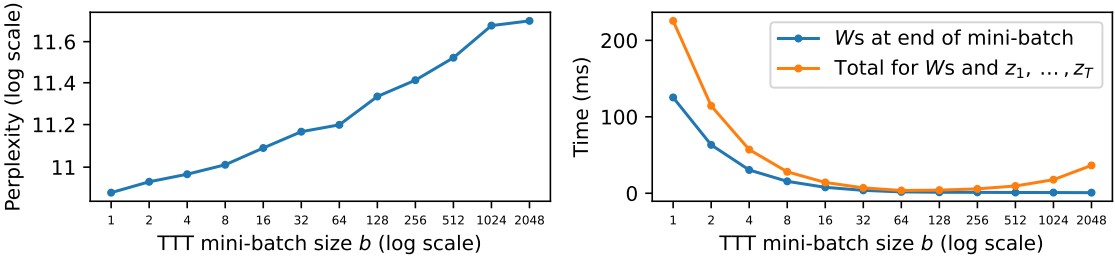

Figure 9. Ablations on TTT mini-batch size $b$, where $b = 1$ is online GD and $b = T$ is batch GD. We choose $b = 16$ for all experiments in this paper. **Left**: Smaller $b$ improves perplexity since more GD steps are taken.[1] The perplexity of 11.09 at $b = 16$ corresponds to the final result of TTT-Linear in Figure 3. **Right**: Forward time in dual form, with context length $T = 2048$. Total time (orange) can be decomposed into time for computing the $W$s at the end of every mini-batch (blue) and time for $z_1, \ldots, z_T$ (orange $-$ blue).[2] Time complexity for the $W$s is $O(T \times d^2)$, constant in $b$, but the blue line decreases as larger $b$ allows more parallelization until hardware utilization saturates. Time complexity for $z_1, \ldots, z_T$ is $O(T \times b \times d)$, so the orange line first decreases with more parallelization, then increases as the extra computation for $z_1, \ldots, z_T$ becomes dominant.

---

[1] In theory, $b$ can potentially be too small such that the variance between mini-batches is too high, hurting optimization. However, we have not observed such an effect in practice.

[2] For Figure 9, we use a single TTT layer in TTT-Linear 1.3B, implemented in pure PyTorch. Our fused kernel significantly improves time efficiency, but makes it difficult to cleanly decompose the time for computing $W_b$ vs. $z_1, \ldots, z_b$.

# A DUAL FORM

The goal of this section is to derive the dual form for general MLPs of arbitrary depth, with nonlinear activations.

Without loss of generality, consider $\eta = 1$ for convenience, and consider only the first mini-batch, where $t = 1, \ldots, b$. Denote:

$$\hat{x}_t = \theta_K x_t, \quad y_t = \theta_V x_t, \quad \bar{x}_t = \theta_Q x_t.$$

Also denote $\hat{X} = [\hat{x}_1, \ldots, \hat{x}_b]$, and $Y$ and $\bar{X}$ analogously. In general, uppercase letters denote matrices whose columns are vectors denoted by the corresponding lowercase letter.

For a network with $K$ layers, denote the initial parameters in layer $k$ by $W_0^k$. Our convention is to use superscripts for the layer and subscripts for time.

## A.1 FORWARD PASS

During the initial forward pass of TTT, we denote the input to layer $k$ by $\hat{X}^k = [\hat{x}_1^k, \ldots, \hat{x}_b^k]$, with $\hat{X}^1 = \hat{X}$. Now we write the forward pass of TTT using these notations.

For $k = 1, \ldots, K$:

- $Z^k = W_0^k \hat{X}^k$

- $\hat{X}^{k+1} = \sigma_k(Z^k)$

where $\sigma_k$ for $k = 1, \ldots, K$ can be any element-wise operation ($\mathbb{R} \mapsto \mathbb{R}$) with derivative $\sigma'$.

Given $\hat{X}^{K+1}$, we compute the loss:

$$l = \frac{1}{2}\ell\left(W_0^1, \ldots, W_0^K; \hat{X}\right) = \frac{1}{2}\left\|\hat{X}^{K+1} - Y\right\|_F^2 = \sum_{t=1}^{b} l_t,$$

where $l_t = \frac{1}{2}\|\hat{x}_t^K - y_t\|^2$ is the same as defined in Equation 2.3, except scaled by $1/2$ for convenience.

All the operations above (except $\sigma$) are `matmuls` and `sums`, therefore are hardware efficient. Both the primal form and the dual form share these initial operations.

## A.2 PRIMAL FORM

The primal form first computes $G_t^k = \nabla_{W_0^k} l_t$ for $t = 1, \ldots, b$, then updates $W_t^k = W_0^k - \sum_{s=1}^{t} G_s^k$. Finally, given $\bar{X}^1 = [\bar{x}_1^1, \ldots, \bar{x}_b^1] = \bar{X}$, the primal form repeats the forward pass with the updated $W$s.

For $k = 1, \ldots, K$:

- $\bar{z}_t^k = W_t^k \bar{x}_t^k$, for $t = 1, \ldots, T$

- $\bar{x}_t^{k+1} = \sigma_k(\bar{z}_t^k)$, for $t = 1, \ldots, T$

where $\bar{X}^{K+1} = [\bar{x}_1^{k+1}, \ldots, \bar{x}_b^{k+1}]$ contains the output tokens.

Note that a standard backward pass only computes the sum of the gradients:

$$\nabla_{W_0^k} l = \sum_{t=1}^{b} \nabla_{W_0^k} l_t = \sum_{t=1}^{b} G_t^k,$$

so the computation of the individual terms in the sum $G_t^k$ for $t = 1, \ldots, b$ cannot be batched together into `matmuls`. Similarly, the forward pass in primal form uses a different $W_t$ for each $\bar{x}_t$, therefore also cannot be batched in the same way as a standard forward pass. These non-standard passes have poor hardware efficiency.

### A.3 DUAL FORM

As discussed in Subsection 2.5, the goal of the dual form is to compute $\bar{X}^{K+1}$ and $W_b^1, \ldots, W_b^K$ with only `matmuls` and light-weight operations such as `sums`, $\sigma$, and $\sigma'$. To achieve this goal, we avoid explicitly computing the intermediate variables: $G_t^k$ and $W_t^k$ for $t = 1, \ldots, b$.

The dual form first computes $\nabla_{\hat{X}^{K+1}} l = \hat{X}^{K+1} - Y$, then takes a standard backward pass.

For $k = K, \ldots, 1$:

- $\nabla_{Z^k} l = \sigma_k' \left( \nabla_{\hat{X}^{k+1}} l \right)$

- $\nabla_{\hat{X}^k} l = \left( W_0^k \right)^T \nabla_{Z^k} l$

- $\nabla_{W_0^k} l = \nabla_{Z^k} l \left( \hat{X}^k \right)^T$

Now we can already compute $W_b^k = W_0^k - \nabla_{W_0^k} l$. To compute the output tokens, we do another forward pass.

For $k = 1, \ldots, K$:

- $\bar{Z}^k = W^k \bar{X}^k - \nabla_{Z^k} l \cdot \mathsf{mask} \left( \left( \hat{X}^k \right)^T \bar{X}^k \right)$

- $\bar{X}^{k+1} = \sigma(\bar{Z}^k)$

By the end of the forward pass, we have computed $\bar{X}^{K+1}$.

While this forward pass is non-standard, it only contains `matmuls`, `sums`, $\sigma$, and `mask`, therefore is efficient like the standard forward pass.

### A.4 DERIVATION

To derive the dual form, we show that:

$$\bar{Z}^k = W^k \bar{X}^k - \nabla_{Z^k} l \cdot \mathsf{mask} \left( \left( \hat{X}^k \right)^T \bar{X}^k \right)$$

is the same as what would be computed in the primal form. Specifically, we show that each column $\bar{z}_t^k$ of $\bar{Z}^k$ in the forward pass of the dual equals to $W_t^k \bar{x}_t^k$ in the forward pass of the primal. We invoke a simple fact.

**Fact 1.** *Define matrices* $A = [a_1, \ldots, a_b]$, $Q = [q_1, \ldots, q_b]$, *and* $V = [v_1, \ldots, v_b]$.[3] *Define* $\hat{v}_t = \sum_{s=1}^{t} a_s^T q_t v_s$, *and* $\hat{V} = [\hat{v}_1, \ldots, \hat{v}_b]$, *then* $\hat{V} = V \cdot \mathsf{mask}(A^T Q)$.

Now plug $A = \hat{X}^k$, $Q = \bar{X}^k$, $V = \nabla_{Z^k} l$, and $\hat{V} = W^k \bar{X}^k - \bar{Z}^k$ into the fact above, we have shown the desired equality.

Note that the $\sigma_k$ and $\sigma_k'$ used above can be extended to arbitrary functions that are not necessarily element-wise operations, including normalization layers. This extension can be achieved through, for example, `vjp` (vector-Jacobian product) in standard libraries for automatic differentiation such as JAX and PyTorch. However, the dual form cannot accelerate operations inside $\sigma$ or its `vjp`.

---

[3]Our matrix $A$ would usually be denoted by $K$ in another context. We use $A$ to avoid confusion with the layer number $K$.

## B  NADARAYA-WATSON ESTIMATOR

**Derivation for the Nadaraya-Watson estimator.**  Throughout this section, we use $\mathbf{x}$ to denote the input token $x$ as a random variable. Our desired output is the corresponding output token, another random variable $\mathbf{z}$. This is formulated as estimating the conditional expectation of $\mathbf{z}$:

$$\mathbb{E}[\mathbf{z}|\mathbf{x} = x] = \int p(z|x)\, z\, dz = \int \frac{p(x,z)}{p(x)}\, z\, dz.$$

Since the true probability distributions $p(x)$ and $p(x,z)$ are unknown, we replace them with their kernel density estimations. Specifically, the kernel density estimation for $p(x)$ is:

$$\hat{p}(x) = \frac{1}{n}\sum_{i=1}^{n}\kappa(x, x_i),$$

where each $x_i$ is a piece of training data in general. (Recall that for our paper, $x_i$ is specifically training data for the inner loop, *i.e.* a token, which matches our notation in the main text.)

For estimating $p(x, y)$, we use the product kernel:

$$\hat{p}(x, z) = \frac{1}{n}\sum_{i=1}^{n}\kappa(x, x_i)\, \kappa'(z, z_i).$$

At first sight, it seems absurd to factor the joint probability into two seemingly independent kernels. But in this case, $\kappa'$ can actually be any $\kappa'_i$ dependent on $x_i$, since it will be integrated out. So the two kernels do not need to be independent.

Plugging in those estimations, we obtain the Nadaraya-Watson estimator:

$$
\begin{aligned}
\hat{\mathbb{E}}[\mathbf{z}|\mathbf{x} = x] &= \int \frac{\hat{p}(x,z)}{\hat{p}(x)}\, z\, dz \\
&= \frac{1}{\hat{p}(x)}\int \hat{p}(x,z)\, z\, dz \\
&= \frac{1}{\sum_{i=1}^{n}\kappa(x, x_i)}\int \sum_{i=1}^{n}\kappa(x, x_i)\, \kappa'(z, z_i)\, z\, dz \\
&= \frac{1}{\sum_{i=1}^{n}\kappa(x, x_i)}\sum_{i=1}^{n}\kappa(x, x_i)\int \kappa'(z, z_i)\, z\, dz \\
&= \frac{1}{\sum_{i=1}^{n}\kappa(x, x_i)}\sum_{i=1}^{n}\kappa(x, x_i)\, z_i.
\end{aligned}
$$

**Asymmetric kernels.**  In modern days, people think of kernels as positive semi-definite, which might not be guaranteed for $\kappa$ unless $\theta_K = \theta_Q$. However, people working on kernels decades ago, around the time when the Nadaraya-Watson estimator was popular, have been very lenient with the choice of kernels, and asymmetric kernels such as our $\kappa$ have enjoyed a long tradition: When a kernel estimator uses $\theta_K \neq \theta_Q$, it is known as a balloon estimator (Chen, 2017). Papers such as Breiman et al. (Breiman et al., 1977) have even used $\theta_Q$ as a function of $x'$, known as sample-adaptive smoothing.

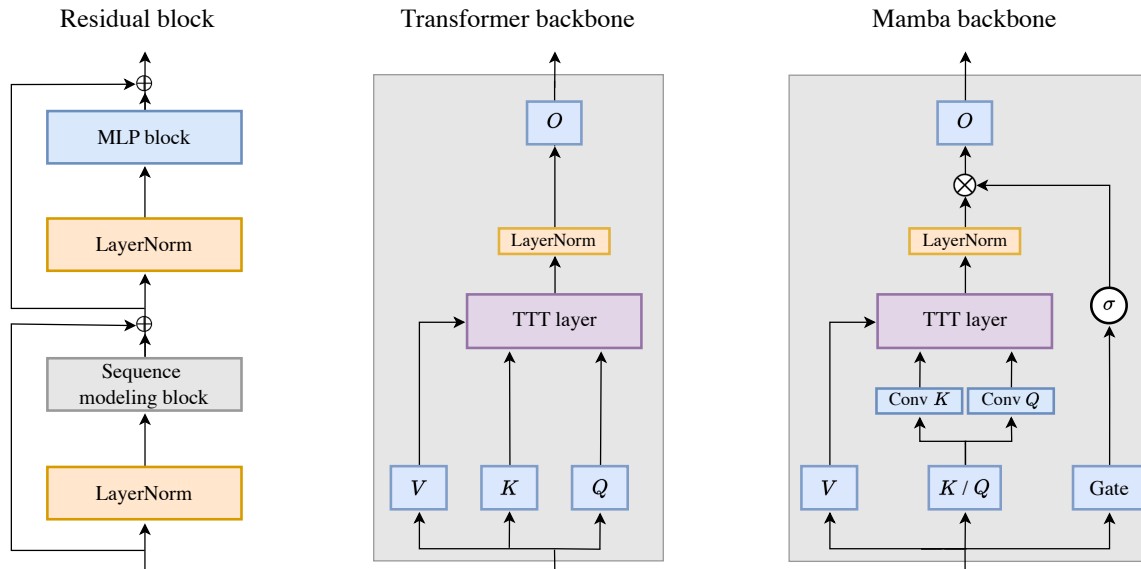

Figure 10. **Left**: A residual block, the basic building block for Transformers. The sequence modeling block is instantiated into two variants: the Transformer backbone and Mamba backbone. **Middle**: TTT layer in the Transformer backbone. The LN before $O$ comes from NormFormer (Shleifer et al., 2021). **Right**: TTT layer in the backbone inspired by Mamba (Gu & Dao, 2023) and Griffin (De et al., 2024). Following these two architectures, $\sigma$ here is GELU (Hendrycks & Gimpel, 2016). To accommodate the extra parameters of the gate without changing the embedding dimension, we simply combine $\theta_K$ and $\theta_Q$ into a single projection.

| Params. | Blocks | Embed. dim. | Heads | Train steps | Peak LR | Tokens |
|---|---|---|---|---|---|---|
| 125M | 12 | 768 | 12 | 4800 | 3e-3 | 2.5B |
| 350M | 24 | 1024 | 16 | 13500 | 1.5e-3 | 7B |
| 760M | 24 | 1536 | 16 | 29000 | 1.25e-3 | 15B |
| 1.3B | 24 | 2048 | 32 | 50000 | 1e-3 | 26B |

Table 2. Training configurations for all experiments. This table reproduces Table 12 in the Mamba paper. The only difference is that the learning rate they use for Mamba and Transformer is $5\times$ the values in their Table 12, and we report the actual values ($5\times$). Note that this table only applies to TTT-Linear, TTT-MLP, and Transformers, as Mamba does not follow the multi-head residual block structure inherited from Transformers.

## C  EXPERIMENT DETAILS

**Architectures.**  Our Transformer strictly follows the construction in the Mamba paper, where *Transformer* is called *Transformer++*. Specifically, the Transformer architecture is based on Llama (Touvron et al., 2023), with rotary positional encodings (RoPE) (Su et al., 2023), SwiGLU MLP blocks (Shazeer, 2020), and RMSNorm (Zhang & Sennrich, 2019) instead of LayerNorm. Our Mamba baseline uses the public code provided by the authors. We have verified that our baselines can reproduce the numbers reported in (Gu & Dao, 2023).

**Training configurations.**  Our training configurations are in Table 2, which simply reproduces Table 12 in the Mamba paper. Following the Mamba paper, we always use 0.5M tokens per training batch regardless of context length. That means for context length $T$ we have 0.5M $/ T$ sequences per batch (assume divisible).

All of our optimization hyper-parameters follow the "improved recipe" in Appendix E.2 of the Mamba paper, reproduced below:

- AdamW optimizer: $\beta = (0.9, 0.95)$

- Cosine schedule: decay to end learning rate $1e - 5$

- Linear learning rate warmup over 10% of the training steps

- Weight decay: 0.1

- Gradient clipping: 1.0

- No Dropout

- Mixed Precision

All models are trained using the Llama tokenizer. For experiments on the Pile, this is the only difference with the recipe in the Mamba paper, which uses two other tokenizers. For experiments on Books, we find that the original angle of the RoPE encoding (Su et al., 2023) $\theta = 10,000$ is sub-optimal for our Transformer baseline in long context. Starting at context length 4k, we try $\theta = 500,000$ following the Llama Long paper (Xiong et al., 2023), and use the better perplexity for Transformer (both pretrain and finetune).

**Transformer finetuning.** Finetuning starts a new cosine schedule with the same optimization hyper-parameter as training from scratch, except the peak learning rate. We try three peak learning rates for finetuning: 1e-5, 1e-4, and 1e-3, and select for the best perplexity. We observe that 1e-4 works the best for the 125M models, while 1e-5 works the best for 350M and larger. This observation is reasonable considering that the end learning rate for the Chinchilla recipe is 1e-5.

**Learning rate for TTT.** As mentioned in Subsection 2.7, the inner-loop base learning rate $\eta_{\text{base}}$ is set to 1 for TTT-Linear and 0.1 for TTT-MLP. Our heuristic for setting $\eta_{\text{base}}$ is similar to how people set the outer-loop learning rate for regular training: We tried $\eta_{\text{base}} \in \{0.01, 0.1, 1, 10\}$ and used the largest value that does not cause instabilities. For TTT-MLP, we use linear warmup for $\eta_{\text{base}}$ over 10% of the training steps, similar to regular training. The number of training steps in the inner loop is $T/b$ (assume divisible). For TTT-Linear, we tried linear warmup in the inner loop but did not observe a difference.

**Experiments in Figure 5.** To ensure fairness to Mamba, all methods in these experiments have matched training FLOPs and are trained with the same recipe (last row of Table 2) as Mamba 1.4B. To match FLOPs with Mamba, Transformer has 19 blocks instead of 24. For TTT-Linear and TTT-MLP, their FLOPs are already close to those of Mamba, so we change the hidden dimension of the MLP blocks from 5504 to 5808 (TTT-Linear) and 5248 (TTT-MLP).

**Gradient checkpointing through time.** By default, libraries such as JAX and PyTorch save the intermediate activations during a forward pass so they can be reused during the backward pass. However, for a TTT layer with $W$ as hidden state, this default saves $W_1, \ldots, W_T$, which uses too much memory. With TTT mini-batch and the dual form, we still need to save (assume divisible) $\kappa = T/b$ $W$s at the end of the mini-batches. A standard technique to save memory in this scenario is gradient checkpointing (Chen et al., 2016), which is usually applied through layers, but we apply it through time.

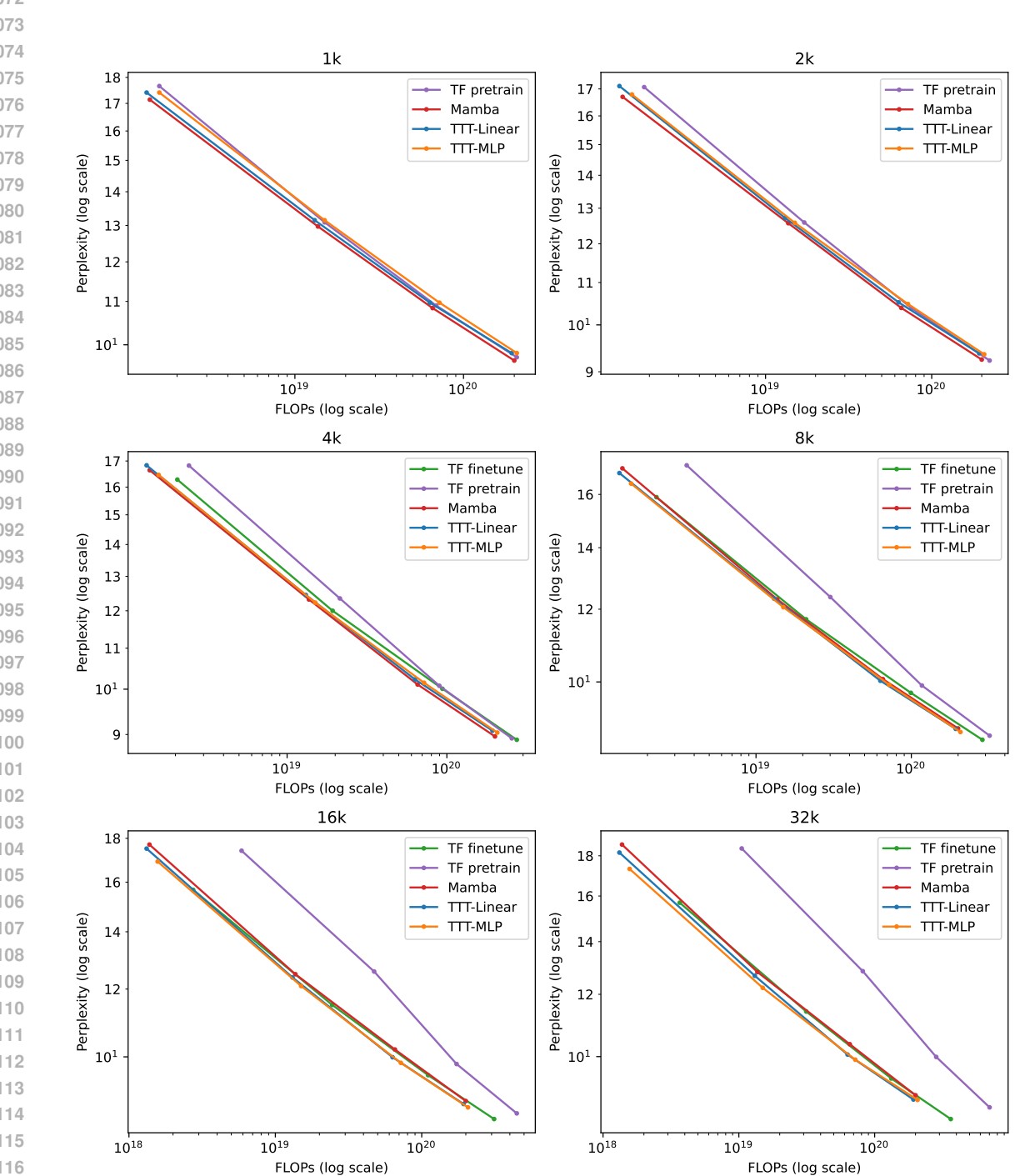

Figure 11. Complete results on Books, presented by context lengths. Figure 4 in Subsection 3.2 presents the subset of results for context lengths 2k and 32k.

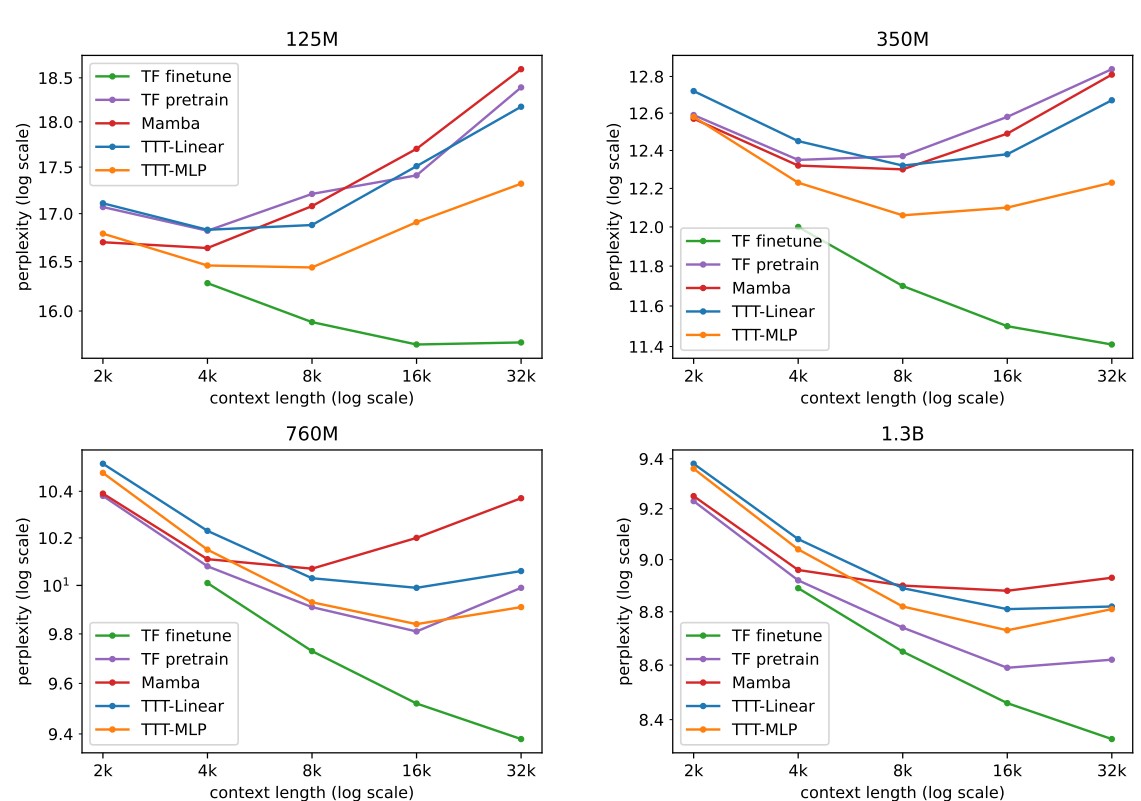

Figure 12. An alternative view of our complete results on Books, presented by model sizes, with context length as the x-axis. For all methods trained from scratch, perplexity becomes worse once the context length becomes too large. This trend is not observed with TF finetune, except for one case at the 125M scale. The best context length increases for larger models (trained from scratch).

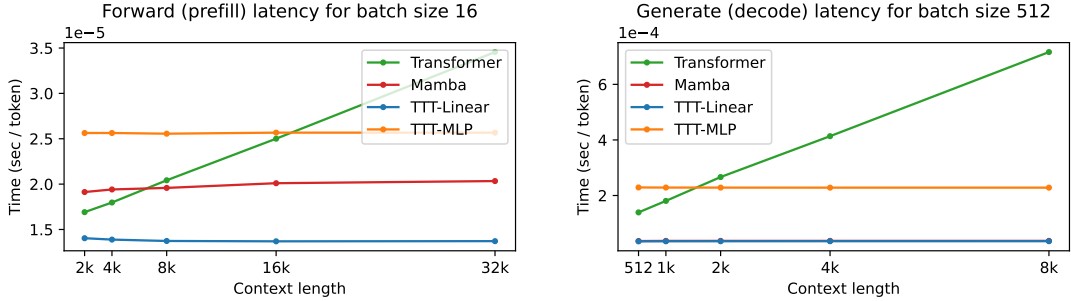

Figure 13. Benchmark on an NVIDIA A100 GPU with 80G HBM and PCIe connections. All models are 1.3B. Time per token grows linearly for Transformer as context length increases, but stays roughly constant for the other methods. **Left**: Forward (prefill) latency for batch size 16. **Right**: Generate (decode) latency for batch size 512. TTT-Linear and Mamba have almost the same latency, which is significantly smaller than that of Transformer and TTT-MLP.

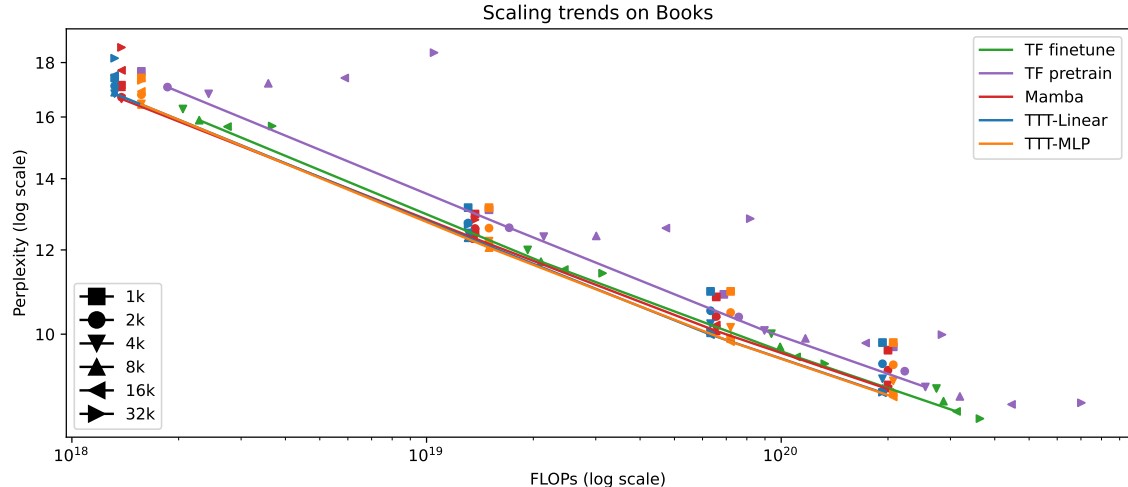

Figure 14. Experiments on Books with context lengths ranging from 1k to 32k. We treat context length as a hyper-parameter and connect the selected points. Since we have Transformers trained from scratch and finetuned, we label them as *TF pretrain* and *TF finetune*. The left panel of Figure 5 is a zoomed-in view between 350M and 1.3B parameters, where *Transformer* is *TF finetune*, the stronger Transformer baseline. For all methods trained from scratch (including TF pretrain), perplexity becomes worse once the context length becomes too large. This trend is highlighted in Figure 12 (in Appendix). We leave further investigation of this trend to future work.

