# OpenReview forum: "Learning to (Learn at Test Time): RNNs with Expressive Hidden States"
_ICLR.cc/2025/Conference — Submitted to ICLR 2025_

### Official Review · Reviewer_ZsP1 · 2024-10-26

**Soundness:** 3
**Presentation:** 3
**Contribution:** 3
**Rating:** 8
**Confidence:** 3

**Summary:**

The authors introduce a new RNN layer they call Test-Time Training (TTT) layer. The hidden state in this layer is a machine learning model itself, in practice a linear layer or an MLP. The update for this layer is a single step of "self-supervised" learning using SGD. The authors show how to efficiently implement their layer on the GPU, or similar hardware. Finally a number of different models using TTT layers are empirically evaluate it terms of perplexity, training FLOPs and inference time on natural language tasks.

**Strengths:**

The paper is well written and easy to follow.

It presents a novel idea which it thoroughly investigates at a number of different scales.

Some of the new models using TTT layers seem to present gains over Mamba and Transformers in terms of training efficiency, and inference latency thought the significance of these gains is questionable.

**Weaknesses:**

The motivation for the self-supervised loss used is a little weak. The self supervised loss described seem to be learning a mapping from one linear projection of a vector to another. I believe this mapping would have a closed from solution as a function of the projections, at least for the linear case. I would like to see the author expand their justification here and if relevant give examples of other inner loop losses experimented with. Self supervised losses for sequence data conventionally try to predict the next item in a sequence from the preceding elements. Did you try this? If so why not?

The efficiency gains are not really put into a human understandable form. Additionally the significance of the efficiency gains seem to be very marginal. While I appreciate multiple runs is likely out of scope for experiments of this size its hard to know if the increases are statistically significant.

It would be great to see if it was indeed the convolution through time that was responsible for the gains when using the Mamba backbone, or if it was some other aspect of the architecture.

The conclusion section is very weak.

**Questions:**

Did you experiment with other inner loop losses? What is the intuition behind the mutli-view loss, was the loss selected to produce Theorem 1?

It does not seem obvious why you'd want to run and iterative optimisation algorithm on the inner loop objective that has a closed form solution, can you give more intuition here?

Did you have any strong justification to think it convolution through time was responsible for the gains when using the Mamba backbone?

I'm not super familiar with the hardware side of LLMs. In section 3.3 you mention wrote two kernels for the TTTlayers. Were these necessary to i) get a fair comparison with the other models? or ii) necessary to make the TTT layers cooperative with other baselines? Without how did the wall clock time compare without these kernels? Can you explain why were these kernel necessary?

---

> ### Author Response · Authors · 2024-11-25
>
> Thank the reviewer for the thoughtful questions. Let’s first address the concerns:
>
> Significance of the efficiency gains: We have further improved the speed of our prefill kernel, as shown in our updated Figure 13 (in Appendix). We hope the differences are now significant :) We have also implemented the training kernels for TTT-Linear and TTT-MLP, as shown in Figure 7 in the revision.
>
> Reconstruction loss has a closed form solution in the linear case: For non-autoregressive problems, yes, but the closed form solution is too compute intensive to be practical, since it needs to solve a linear system of size d-by-d. For autoregressive problems, the hidden state (i.e. the set of parameters of the inner model) needs to be different at each timestep. Unfortunately, the closed form solution only produces a single set of parameters.
>
> Next-token prediction as self-supervised loss: Yes, we have tried next-token prediction at small scale, the result is very slightly worse, changing from 11.09 (as reported in Table 1) to 11.13.
>
> Was convolution through time responsible for gains in the Mamba backbone: Yes. The two main changes in the backbone are: 1) convolutions, 2) SwiGLU around the sequence modeling layer. If we only add SwiGLU, perplexity actually gets worse, changing from 11.99 (in Table 1) to 12.47. Since the backbone is inherited from Mamba and orthogonal to our contributions, we have not investigated further why convolutions are important to the backbone.
>
> Conclusion section is weak: We have added a richer conclusion section in the revision.
>
> We believe that the reviewer’s first three questions have already been answered as we addressed the concerns. For the reviewer’s last question about the kernels, yes, they are necessary for fair comparison. Every widely used sequence modeling layer has been written into GPU kernels, including Mamba and self-attention. Transformers without their custom kernels for self-attention would likely be an order of magnitude slower, because most of the time would be spent on memory I/O. The kernels allow the programmers to control how low-level memory is used on GPUs.

---

> > ### Comment · Reviewer_ZsP1 · 2024-11-28
> > **Reconstruction loss**
> >
> > I am still slightly confused by the framing of the reconstruction loss. As you said the inner minimisation has a closed form for the linear case. However, $W_0$ is also learnt so could just learn to be this value? In terms of the inner optimisation framing this would be optimal, but I believe this would not lead to a good model over all? Your comment that "For autoregressive problems, the hidden state (i.e. the set of parameters of the inner model) needs to be different at each time step" reinforces this idea. Personally I feel I am lacking intuition behind the multi-view loss. Was it chosen in order to derive a linear attention like update?

---

> > > ### Author Response · Authors · 2024-11-29
> > >
> > > Thank the reviewer for participating in discussion. Answering the questions:
> > >
> > > >As you said the inner minimisation has a closed form for the linear case. However, $W_0$ is also learnt so could just learn to be this value? In terms of the inner optimisation framing this would be optimal, but I believe this would not lead to a good model over all?
> > >
> > > The optimization problem is given by the test sequence. $W_0$ is learned only on the training set. It does not know this optimization problem, therefore cannot learn to be the optimal value. It can only learn a good initialization on average, across the distribution of sequences.
> > >
> > > >Personally I feel I am lacking intuition behind the multi-view loss. Was it chosen in order to derive a linear attention like update?
> > >
> > > We would still have constructed a multi-view reconstruction loss even without knowing linear attention. In fact, we have been working along this line for years without realizing the connection. However, the specific parameterization of the views is indeed chosen to derive an attention-like update.

---

### Official Review · Reviewer_HodN · 2024-11-04

**Soundness:** 3
**Presentation:** 2
**Contribution:** 3
**Rating:** 5
**Confidence:** 4

**Summary:**

The paper proposes Test-Time Training (TTT) layers. TTT outperforms Transformers and Mamba for models up to 1.3B parameters. In terms of runtime, TTT-Linear is more efficient than Transfromers and is competitive with that of Mamba.

**Strengths:**

This paper explores a gradient-based alternative to modern RNNs and self-attention. The approach is uncommon and allows for a more expressive hidden state, which results in better performances.

TTTs achieve competitive or slightly better performance compared to Mamba and Transformers.

Connecting TTT with linear and self attention is a nice touch.

**Weaknesses:**

Mamba is parallelized with a parallel scan. However, TTT requires some sequential computations controlled by a hyperparameter $b$, potentially making it significantly slower during training. Unfortunately, the paper lacks memory and runtime analyses during training, making it rather hard to determine the practicality of training a TTT model compared to other modern RNNs that leverage parallel scan (e.g., Mamba, xLSTM, GLA-Transformers, etc...).

Meta-learning is closely related to TTT, but the connection is not properly discussed. The paper briefly mentions a connection on page 3, but the discussion is lacking. For example, TTT using "online gradient descent" is reminiscent of MAML, where the task is a sequence and each sample is a timestamp.

When initially reading, the reviewer found the usage of the terms "batch GD" and "mini-batch GD" to be confusing and took some time to get used to. For example, when mentioning "mini-batch GD", the reviewer initially thought of its typical use case (per outer-loop training iteration). However, in this work, the mini-batch GD is done temporally with varying timestamps.

**Questions:**

In meta-learning, a common issue of the inner/outer loop formulation with gradient descent is the second-order gradients which are memory intensive. How does TTT resolve this issue?

Could the authors distinguish their usage of (mini-)batch GD from the commonly associated connotation? For example, prefacing the term with "temporal" could potentially reduce the initial confusion.

The training efficiency of TTT appears to be highly dependent on $b$ as it determines the amount of parallelism TTT can leverage. Furthermore, $b$ also significantly affects the performance. In contrast, methods such as Mamba are not dependent on this hyperparameter. Plots comparing the training runtime, memory, and performance of TTT with varying values of $b$ and methods such as Mamba are important. Could the authors include these?

---

> ### Author Response · Authors · 2024-11-25
>
> HodN
>
> Thank the reviewer for the detailed review. Addressing the reviewer’s concerns:
>
> Memory and runtime analysis during training:
> We have recently finished implementing the training kernels for TTT-Linear and TTT-MLP. Please refer to Figure 7 in the revision for a plot of training latency. A 1B TTT-Linear model is 2x faster than Mamba, and 1B TTT-MLP model is only 20% slower. In terms of memory usage (GPU DRAM), our maximum batch size per device during training is the same as that of Mamba. This is reasonable since the outer loop parameters take up most of the memory.
>
> Connection to meta-learning:
> We have added a comprehensive discussion in the revision.
>
> Second-order gradients are memory intensive:
> We train TTT layers in dual form, as explained in Subsection 2.5. So we do not have second-order gradients. One of our main technical contributions is deriving the dual form for TTT layers with any neural network as inner model.
>
> Plots comparing the performance and runtime for various values of TTT mini-batch size b:
> We have already provided this comparison in Figure 9 in the Appendix. Sorry that we could not find enough space to include it in the main text.
>
> Overall:
> We are glad that the reviewer found our paper “good” in contributions (3 out of 4), and “fair” in soundness and presentation (2 out of 4). Have we been able to adequately address the reviewer concerns? If so, would the reviewer please kindly consider adjusting the score?

---

> > ### Comment · Reviewer_HodN · 2024-11-26
> >
> > Thank you for your response. Some of my concerns have been addressed, and as a result, I have adjusted my score accordingly.
> >
> > However, I believe further analysis is needed regarding the hyperparameter $b$ and its impact on the practicality of the method. Figure 9 currently compares only TTT's performance and runtime with respect to  $b$, but it does not include comparisons with the existing baseline methods in the paper (Transformers or Mamba). Additionally, Figure 9 is missing a plot for training memory usage. Without these comparisons, it is difficult to assess, given a fixed computational budget, whether one should prefer TTT over prior methods.
> >
> > For instance, the y-axis could include dotted lines representing existing baseline methods like Mamba, showing their respective performance (e.g., perplexity), runtime (ms), and memory usage. This would provide clearer context for evaluating TTT's relative advantages or trade-offs.

---

> ### Author Response · Authors · 2024-11-27
>
> Thank the reviewer for participating in discussion. Addressing the additional concerns:
>
> >“However, I believe further analysis is needed regarding the hyperparameter $b$ and its impact on the practicality of the method.”
>
> A method is practical as long as any one set of its hyper-parameters produces useful results across a variety of experiments. Knowing the results for other non-default sets of hyper-parameters can help with understanding the method, but is not necessary to determine practicality. Throughout all of our experiments (except those in Figure 9), we have held the inner-loop mini-batch size $b=16$ constant. So the practicality of our method should be determined according to the results for $b=16$.
>
> >“Figure 9 currently compares only TTT's performance and runtime with respect to $b$, but it does not include comparisons with the existing baseline methods in the paper (Transformers or Mamba) ...... For instance, the y-axis could include dotted lines representing existing baseline methods like Mamba, showing their respective performance (e.g., perplexity), runtime (ms), and memory usage.”
>
> The inner-loop mini-batch size $b$ is particular to the TTT approach. Transformers and Mamba do not have this hyper-parameter, so we cannot add a non-constant line for these methods to Figure 9. As discussed above, we keep $b=16$ for all experiments. Results in Section 3, such as Figure 3 and 4, compare TTT ($b=16$) with Transformers and Mamba in perplexity. Figure 6 compares them in runtime.
>
> >“Additionally, Figure 9 is missing a plot for training memory usage.”
>
> Here is the memory usage in GB of the various methods at 125M scale, on an A100 with 80GB memory, with context length 2048, for various outer-loop mini-batch sizes. The memory footprint of TTT is comparable to Mamba and the most optimized Transformer (with FlashAttention). Sorry we cannot obtain memory usage for every inner-loop mini-batch size $b$, because our kernel is specifically written for $b=16$. As discussed above, as long as memory usage is reasonable for $b=16$, our method should be practical in this aspect.
>
> | Batch Size | Transformer | Mamba | TTT-Linear | TTT-MLP |
> |------------|-------------|-------|------------|---------|
> | 1          | 4.1         | 4.0   | 4.2        | 4.5     |
> | 2          | 6.1         | 5.9   | 6.4        | 7.0     |
> | 4          | 10.3        | 9.7   | 10.8       | 11.8    |
> | 8          | 18.5        | 17.3  | 19.5       | 21.6    |
> | 16         | 34.9        | 32.4  | 37.0       | 41.0    |
> | 32         | 67.9        | 62.7  | 70.9       | 77.6    |
>
> Have we been able to adequately address your concerns? Thank you again for your participation.

---

### Official Review · Reviewer_eEfe · 2024-11-06

**Soundness:** 2
**Presentation:** 2
**Contribution:** 2
**Rating:** 6
**Confidence:** 3

**Summary:**

This paper introduces Test-Time Training (TTT) layers, which make the hidden state a model updated via self-supervised learning, enabling efficient long-context processing with linear complexity. Two implementations, TTT-Linear and TTT-MLP, outperform Transformers and Mamba on long-context tasks.

**Strengths:**

TTT layers offer efficient long-context processing with linear complexity, outperforming traditional Transformers and Mamba in terms of scalability and performance.

**Weaknesses:**

The paper shows limited improvement over Mamba, and its evaluation is restricted to language modeling tasks. Furthermore, it lacks comparative experiments with related methods like Fast Weight Layers and Hidden-State Optimization.

Reconsidering the Past: Optimizing Hidden States in Language Models
Davis Yoshida, Kevin Gimpel, EMNLP2021 findings
Meta-Learning Fast Weight Language Models
Kevin Clark, Kelvin Guu, Ming-Wei Chang, Panupong Pasupat, Geoffrey Hinton, Mohammad Norouzi, EMNLP2022

**Questions:**

Q1. Could you demonstrate statistically significant performance improvement by adding error bars to the experimental results? While TTT-Linear shows similar computational efficiency to Mamba, its perplexity appears nearly identical. Without error bars in the results, it is difficult to determine how effective this method truly is.

Q2. Is it possible to include evaluations on downstream tasks as part of the experiment?
In Mamba’s original paper, experiments were conducted on downstream tasks to demonstrate performance beyond language modeling alone. At a minimum, it seems necessary to include experiments similar to those in Section 4.2.2 of the following paper:
https://arxiv.org/abs/2312.00752

Q3. Should comparative experiments with the following studies also be conducted? Also,one seems to be a missing citation.

1. Reconsidering the Past: Optimizing Hidden States in Language Models, Davis Yoshida, Kevin Gimpel, EMNLP 2021 Findings
2. Meta-Learning Fast Weight Language Models, Kevin Clark, Kelvin Guu, Ming-Wei Chang, Panupong Pasupat, Geoffrey Hinton, Mohammad Norouzi, EMNLP 2022

---

> ### Author Response · Authors · 2024-11-25
>
> Thank the reviewer for the detailed review. Addressing the reviewer’s concerns:
>
> Adding error bars: We now have error bars for our largest experiment: 1.3B models with 32k context length on Books. We did 4 additional training runs (5 total) for each method, randomizing over model initialization and data ordering. The standard deviation is 0.01 across all methods. Specifically, the perplexities are
> TTT-Linear	8.82 ± 0.01
> TTT-MLP 	8.81 ± 0.01
> Mamba 	8.93 ± 0.01
> Sorry that we could not obtain error bars for all experiments due to the large cost of training LLMs. We also had to exclude Transformer in the runs above due to the prohibitive cost of training at 32k context.
>
> Evaluation on downstream tasks: The focus of our paper is on long context. However, all the tasks in 4.2.2 of Mamba (Gu and Dao, 2024), as the reviewer suggested, have sequence length <1000. In fact, most of them only need ~100 tokens. There also exist downstream tasks with long context, such as book summarization and solving software issues in large repositories. However, these tasks require post-training for instruction following capability, and pre-training frontier models (usually > 3B) with trillions of tokens. Unfortunately, the cost of this evaluation exceeds our academic budget by at least 500x.
>
> Comparison with Clark et al. 2022 and Yoshida et al. 2021: The focus of our paper is on long context, specifically models with linear complexity. The two papers above also explore the idea of training at test time, but still have the quadratic complexity of Transformers, therefore are out-of-scope for our evaluations. Meta-Learning Fast Weight Language Models (Clark et al. 2022) adds a single fast weights layer on top of a Transformer. Optimizing Hidden States in Language Models (Yoshida et al. 2021) optimizes the entire KV cache of a Transformer. We have added citation for Yoshida et al. 2021 in our related work section.

---

> > ### Comment · Reviewer_eEfe · 2024-11-30
> >
> > Thank you for your response which is reasonable, so I have raised the score.

---

### Official Review · Reviewer_o5Ft · 2024-11-06

**Soundness:** 1
**Presentation:** 2
**Contribution:** 1
**Rating:** 5
**Confidence:** 4

**Summary:**

This paper introduces sequence modeling layers called Test-Time Training (TTT) layers. The key contributions are:
1. Introduction of a machine learning model (e.g., linear model or MLP) that is trained through self-supervised learning during inference/test time. This allows the model to learn and adapt to patterns in the input sequence on the fly.
2. Two practical instantiations: TTT-Linear and TTT-MLP, which use a linear model and a two-layer MLP respectively as their hidden states.
3. Empirical validation showing that:
    * Both TTT variants match or exceed performance of Transformers and Mamba models across different model sizes (125M to 1.3B parameters)
    * TTT-Linear achieves faster inference than Transformers at 8k context and matches Mamba's speed
The authors also introduce practical optimizations like mini-batch TTT and a dual form implementation that significantly improve wall-clock time performance, making TTT-Linear immediately practical for real applications.

**Strengths:**

The paper is tackles an interesting problem but lacks novelty and experimental rigour.

**Weaknesses:**

The major weakness of the paper is the missing novelty, as well as comparison to appropriate baselines, in my opinion.

* Novelty: Linear self-attention and its variants are clearly motivated already by learning at inference. This is discussed in e.g.  Schlag et al., 2021 (Linear Transformers Are Secretly Fast Weight Programmers) and citations within. Additionally, and building on Schlag, are at least 3 missing citations i.e. Yang et al. 2024 (Parallelizing Linear Transformers with the Delta Rule over Sequence Length) as well as von Oswald et al. 2022, (Transformers Learn In-Context by Gradient Descent) and von Oswald et al. 2023 (Uncovering mesa-optimization algorithms in Transformers) which also build on this framework.

* Please provide a comprehensive analysis how TTT-linear differs from these layers.

* Yang et al. here is therefore also one obvious baseline, which is missing. First, the parallelized delta net implementation has better throughput as well as perplexity scores than Mamba and is therefore a crucial and missing baseline here. Additionally, and probably  even closer baseline are results of gated linear self-attention layers e.g. Yang et al. 2023 (Gated Linear Attention Transformers with Hardware-Efficient Training), which seem to be an extension of TTT-linear, because of the gates.

* These missing references also highlight that the following statements might be untrue:
	1) The “novel” class of layers is in my opinion a false statement and needs correction.
	2) Related literature, see references, is able to parallelize across time (gated) linear self-attention and the delta rule, showing that they are able, to an extent, to leverage matmul units on GPUs (and TPUs). What is the relationship to Mini-Batch TTT, is this identical?



* Further things:

	- Figure 3: 1) Perplexity seems very low here on the Pile. 2) Perplexity should be lower for 8k than for 2k, especially for Transformers.
	- Missing evals: Please Incorporate evals such as Table 1 and 2 in  Yang et al. 2024 (Parallelizing Linear Transformers with the Delta Rule over Sequence Length)

**Questions:**

Please see weakness

---

> ### Author Response · Authors · 2024-11-25
>
> Thank the reviewer for the critical feedback.
>
> There are three levels of ideas in our paper:
> 1. Learning at test time and meta-learning at training time.
> 2. A practical framework to do the above with any neural network as inner model.
> 3. TTT-Linear and TTT-MLP as instantiations.
>
> Our claim of novelty is at level 2. All the reviewers, including Reviewer o5Ft, have acknowledged this claim. However, the reviewer’s main concern is lack of novelty at levels 1 and 3. We are confused by this concern because it diverges from our main claim.
>
> Specifically, Reviewer o5Ft has implied that TTT-Linear is similar to DeltaNet (Yang et al. 2024). We completely agree. But TTT with linear model - the most degenerate instantiation - is merely a pedagogical data point. How should people deal with longer context that outgrows the capacity of a linear hidden state? This question is what our paper really tries to answer. For DeltaNet, this question is out of scope.
>
> At the very least, demanding experimental comparisons with DeltaNet seems unfair because it is concurrent work. From the perspective of traditional conference cycles, DeltaNet has not yet been published (accepted to NeurIPS this December). From the perspective of arxiv, our paper was released less than a month after DeltaNet. In fact, our first version was released in 2023.
>
> To be honest, we find zero-sum debates of academic priority harmful to the community, especially when none of the RNNs today, including ours, has truly succeeded in long context. DeltaNet is a very valuable paper. It shows how to extract the maximum potential out of a linear hidden state. We have revised our related work section to reflect that. But our paper is also valuable. It shows a practical path beyond linear hidden states.
>
> For the minor questions, sorry that we are also confused. What’s the problem that perplexity “seems very low here on the Pile.”? And perplexity is indeed lower for 8k than 2k in most cases.

---

> > ### Comment · Reviewer_o5Ft · 2024-11-27
> >
> > Dear authors, Dear Songlin
> >
> > Thank you for your response and comments.
> > I agree that the discussed instantiations of RNNs is valuable research, especially when tackling long memory retention.
> > Nevertheless, for example it is written in the abstract that the paper presents “a new class of sequence modeling layers with linear complexity and an expressive hidden state.”
> > This is not in line with what you now claim your contribution to be, i.e. “2. A practical framework to do the above (Learning at test time and meta-learning at training time) with any neural network as inner model.”
> >
> > If the practical implementation is at the heart of the paper, as the authors say in the rebuttal, I think this should be expressed as such - in which case I would agree the paper could be a valuable contribution to ICLR or any other venue.
> >
> > In terms of missing evaluation, what I meant is a comparison of your layer with the DeltaLayer which is not concurrent work.
> > The main novelty is the TTT-MLP layer (and slight modifications to the DeltaNet as mentioned by Songlin) which seems to perform on par with TTT-Linear (which is very close architecturally to the DeltaLayer, as agreed by all here).
> > So performance wise / architecturally, your paper might show improvements on the DeltaLayer, which not shown in detail at scale or on benchmarks, but otherwise does not improve on it with novel variants i.e. TTT-MLP.
> >
> > This is why I do think the paper is overselling the novelty, and would need a comprehensive rewrite in order to reflect the contributions.
> > So I will keep my score.

---

> > > ### Author Response · Authors · 2024-11-28
> > >
> > > Thank the reviewer for participating in discussion. Addressing the concerns:
> > >
> > > >…it is written in the abstract that the paper presents “a new class of sequence modeling layers with linear complexity and an expressive hidden state.” This is not in line with what you now claim your contribution to be… If the practical implementation is at the heart of the paper, as the authors say in the rebuttal, I think this should be expressed as such - in which case I would agree the paper could be a valuable contribution to ICLR or any other venue.
> > >
> > > We have revised our abstract and introduction (in red). We have put in our best effort to express that our contribution is limited to the practical framework. Does the reviewer feel that we have adequately expressed this limitation? If so, could our paper be a valuable contribution to ICLR?
> > >
> > > >In terms of missing evaluation, what I meant is a comparison of your layer with the DeltaLayer which is not concurrent work.
> > >
> > > We are not sure what the reviewer means by “DeltaLayer”. Which specific prior work is the reviewer referring to?

---

> > > > ### Comment · Reviewer_o5Ft · 2024-12-02
> > > >
> > > > > We are not sure what the reviewer means by “DeltaLayer.” Which specific prior work is the reviewer referring to?
> > > >
> > > > I apologize for the lack of clarity. Here, I specifically refer to the layer introduced by Schlag et al. (2021) in "Linear Transformers Are Secretly Fast Weight Programmers," which employs the delta update rule.
> > > >
> > > > > Does the reviewer feel that we have adequately expressed this limitation? If so, could our paper be a valuable contribution to ICLR?
> > > >
> > > > To reiterate in light of the above clarification: the work by Schlag et al. is not concurrent with yours, but it is functionally very similar to the TTT-Linear layer you propose. As the authors themselves noted, if TTT-Linear is "merely a pedagogical data point," I would expect a more convincing empirical investigation across additional benchmarks to demonstrate how extending the inner model to arbitrary neural (non-linear) networks could improve downstream performance compared to the layer by Schlag et al.
> > > > With a more thorough empirical evaluation and comparison to relevant baselines to establish the practical relevance of the proposed implementation, this work could indeed make a valuable contribution to ICLR. However, in its current state, I believe the paper falls short of the standard required for publication.

---

> ### Public Comment · ~Songlin_Yang1 · 2024-11-26
>
> Dear authors and reviewer:
>
> As the first author of the parallel DeltaNet paper (Yang et al., 2024), I would like to offer some constructive feedback on the review. I find TTT to be a valuable contribution with an interesting perspective and appreciate the authors' inclusion of TTT-linear and DeltaNet comparisons in the updated draft. Notably, TTT-linear with LayerNorm becomes a nonlinear RNN and is no longer equivalent to DeltaNet. As shown in Table 1, this nonlinear enhancement (through LayerNorm and residual connections in f) leads to substantial performance improvements. The authors' proposed hybrid approach – combining intra-chunk linear with inter-chunk nonlinear training strategies – offers an interesting middle ground for studying nonlinear RNNs while maintaining hardware efficiency.
>
> To strengthen the paper further, I suggest:
>
> - Expanding the discussion of related work from meta-learning and fast-weight programming literature, particularly in relation to the NeurIPS 2019 paper "Metalearned Neural Memory" (Sections 3.3-3.4)
>
> - Including evaluations beyond language modeling perplexity to demonstrate broader applicability
>
> - Adding direct experimental comparisons with DeltaNet in future work
>
> These additions would help readers better understand TTT's contributions within the broader context of efficient RNN architectures.
>
>
> Best,
>
> Songlin

---

> ### Author Response · Authors · 2024-12-03
>
> Schlag et al. (2021) is completely sequential across the time dimension. The chunk-wise parallelization in Yang et al. (2024) is necessary for the Delta rule to be practical in large models. To further illustrate our point, we compare the training throughput of a 1.3B model with 8k context using
> - Schlag et al. (2021): 15042 tokens / sec
> - TTT-Linear: 48545 tokens / sec
>
> Schlag et al (2021) is 3x slower than TTT-Linear.
>
> In terms of perplexity, we ran the DeltaNet in Schlag et al. (2021) at 2k context on the Pile, in exactly the same setting as Table 1 in our submission. This model produced NaN even after we have tried a few sets of hyper-parameters. Note that the largest experiment in Schlag et al. (2021) is on WikiText-103 with only 103M words in total. It is common for methods that work in toy settings to be unstable in larger settings. Yang et al. (2021) has significantly modified Schlag et al. (2021) to make it stable in their experiments.

---

> > ### Comment · Reviewer_o5Ft · 2024-12-03
> >
> > I thank the authors for their clarification.
> >
> > In my opinion, the paper is still below the threshold for acceptance, as in my opinion the experiments in this work have failed to convince me of the practical relevance of the proposed method, due in part to a lack of relevant baselines (which I understand is concurrent work), or to its incremental improvement on the available baselines.
> >
> > However, I will raise my score to 5, as I believe the practical framwork is nevertheless of interest to the community.

---

### Meta-Review · Area_Chair_mcjA · 2024-12-21

**Metareview:**

The paper introduces a practical framework for sequence modeling by introducing test-time training (TTT) layers. Two specific instantiations get introduced and are empirically compared to transformers and Mamba. The strengths that reviewers identified are that the paper is well written, the empirical performs seems to be somewhat improved, and that the long-context processing with linear complexity is valuable. On the flip side, several concerns arose over the novelty of the methodology, a potential lack of comparison/relation to several prior similar works, more analysis required wrt e.g. runtime, memory, and statistical significance of results, and marginal gains. Some of these concerns have been partially tackled during the discussion period, but a part of the reviewers remain unconvinced that this is sufficient for paper acceptance. The AC has taken a look at the implemented changes and finds that the additions are a preliminary first step. The added conclusions are more of a bullet point list, the added discussion wrt relation to meta-learning is appreciated, yet a more thorough embedding in the literature and comparison is still open. The AC tends to agree with the reviewers who are worried that the results present marginal improvements at best. As such, the AC unfortunately recommends to reject the paper and suggest that the thorough reviewer feedback is taken into account in a more extensive revision.

Update by SAC: I agree with the recommendation.

**Additional Comments On Reviewer Discussion:**

The discussion with reviewers was very two-sided. On the one hand, reviewer o5Ft engaged in an extensive discussion on the position of the paper, its comparisons, and discussion of related work. The reviewer suggested the novelty is insufficient and even after the rebuttal remains largely unconvinced that the demonstrated empirical gains are enough of an improvement. In similar spirit reviewer HodN has raised their initial score, but remains very borderline due to remaining concerns over insufficiently extensive experimentation, e.g. with respect to hyper-parameters. The only very positive reviewer, reviewer ZsP1, has indicated a very good score, yet the number of mentioned positives remain limited. In contrast, there are some concerns that are shared with other reviewers, such as the fact that efficiency gains are not presented in a sufficiently human understandable form and lack proper corroboration. Other concerns over justification were also added. Following these points, the AC believes that although the subjective scoring of reviewers differs, there are parts that are mutually agreed upon in terms of required improvements for the paper to be published.

---

### Decision · Program_Chairs · 2025-01-22

Reject